# Factors Required for Adhesion of *Salmonella enterica* Serovar Typhimurium to *Lactuca sativa* (Lettuce)

Laura Elpers,[a] Lena Lüken,[a] Fabio Lange,[a] Michael Hensel[a,b]

[a]Abt. Mikrobiologie, Universität Osnabrück, Osnabrück, Germany
[b]Center for Cellular Nanoanalytics (CellNanOs), Universität Osnabrück, Osnabrück, Germany

**ABSTRACT** *Salmonella enterica* serovar Typhimurium is a major cause of foodborne gastroenteritis. Recent outbreaks of infections by *S. enterica* serovar Typhimurium are often associated with non-animal-related food, i.e., vegetables, fruits, herbs, sprouts, and nuts. One main problem related to the consumption of fresh produce is the minimal processing, especially for leafy green salads. In this study, we focused on butterhead lettuce (*Lactuca sativa*) to which *S. enterica* serovar Typhimurium adheres at higher rates compared to *Valerianella locusta*, resulting in prolonged persistence. Here, we systematically analyzed factors contributing to adhesion of *S. enterica* serovar Typhimurium to *L. sativa* leaves. Application of a reductionist, synthetic approach, including the controlled surface expression of specific adhesive structures of *S. enterica* serovar Typhimurium, one at a time, enabled the identification of relevant fimbrial and nonfimbrial adhesins, the O-antigen of lipopolysaccharide, the flagella, and chemotaxis being involved in binding to *L. sativa* leaves. The analyses revealed contributions of Lpf fimbriae, Sti fimbriae, autotransported adhesin MisL, T1SS-secreted BapA, intact lipopolysaccharide (LPS), and flagella-mediated motility to adhesion of *S. enterica* serovar Typhimurium to *L. sativa* leaves. In addition, we identified BapA as a potential adhesin involved in binding to *V. locusta* and *L. sativa* leaf surfaces.

**IMPORTANCE** The number of produce-associated outbreaks by gastrointestinal pathogens is increasing and underlines the relevance to human health. The mechanisms involved in the colonization of, persistence on, and transmission by, fresh produce are poorly understood. Here, we investigated the contribution of adhesive factors of *S. enterica* serovar Typhimurium in the initial phase of plant colonization, i.e., the binding to the plant surface. We used the previously established reductionist, synthetic approach to identify factors that contribute to the surface binding of *S. enterica* serovar Typhimurium to leaves of *L. sativa* by expressing all known adhesive structures by remote control expression system.

**KEYWORDS** adhesome, *Salmonella enterica* serovar Typhimurium, lettuce, *Lactuca sativa*, Lpf fimbriae, Sti fimbriae, MisL, BapA, LPS, flagella-mediated motility, lipopolysaccharide

*S*almonella enterica is a common cause of foodborne gastroenteritis leading to thousands of fatal cases worldwide (1). Currently, the number of *S. enterica* outbreaks associated with the consumption of fresh produce is increasing (2, 3). Besides products of animal origin, *S. enterica* can contaminate fresh produce (4). Experimental studies showed that contamination can occur during preharvest by the seeds themselves (5), irrigation water, animal intrusion (e.g., fecal contamination), or fertilizers (often based on animal origin). After harvest, fresh produce may be contaminated by improper implemented hygiene measures or additional processing steps (washing water, cutting) (6, 7). Further investigations showed that after first attachment and adhesion, *S. enterica* can colonize and persist on and in plants (8–10).

Several studies investigated the interaction of *S. enterica* and various leafy green salad species by focusing on the important first contact of *S. enterica* with leafy green salad. It was

Address correspondence to Michael Hensel, Michael.Hensel@uni-osnabrueck.de.

The authors declare no conflict of interest.

shown that *S. enterica* serovar Thompson adheres to spinach leaves in higher numbers than watercress leaves. Attachment to abaxial and adaxial leaf surfaces was comparable (11). Differences in levels of attachment to various leafy green salad species were further investigated for *S. enterica* serovar Typhimurium (*S.* Typhimurium). Adhesion of *S.* Typhimurium to *L. sativa* leaves (lettuce) occurred in higher numbers than *V. locusta* leaves (corn salad, mâché, lamb's lettuce); moreover, persistence *in planta* was longer for *L. sativa* than for *V. locusta* (12).

For the adhesion of *S. enterica* to leafy green salad, mostly, the involvement of the flagella has been investigated so far. Directed motility is involved in adhesion to iceberg lettuce leaves, while loss of flagella filaments ablated binding of *S.* Typhimurium. The binding of a smooth-swimming Δ*cheY* strain was not affected, whereas the internalization through stomata was prevented (13). Other studies addressed the involvement of genes in long-term persistence on lettuce leaves under cold storage. Previous work (14) revealed a possible involvement of *bcsA*, *misL*, and *yidR* due to decreased attachment to lettuce leaves and decreased survival during cold storage for respective mutant strains.

Investigations of *Salmonella*-plant interactions are complex since different sites of contamination (roots, leaves, fruits, seeds), age of leaves, metabolic changes of plants during dark-night rhythm, various temperatures, and other environmental stresses (e.g., UV light) have to be considered. Here, we focus on the adhesion of *S.* Typhimurium to butterhead lettuce (*Lactuca sativa*) leaves and aim to reveal factors involved in the initial attachment of *S.* Typhimurium. While previous studies mainly investigated *S.* Typhimurium wild type (WT) or strains mutated in candidate adhesion factors, we focus on the expression of defined adhesive structures, one at a time, to reveal possible ligands on *L. sativa* leaf surfaces.

*S.* Typhimurium possesses a complex set of adhesive structures, including 12 chaperone-usher (C/U) fimbriae, curli fimbriae assembled by the nucleation-precipitation pathway, 2 type 1 secretion system (T1SS)-secreted adhesins (BapA, SiiE), 3 type 5 secretion system (T5SS)-secreted (autotransported) adhesins (MisL, ShdA, SadA), and 2 outer membrane proteins (OMPs) with putative adhesive features (PagN and Rck) (15, 16). For most C/U fimbriae, little is known about the conditions of native expression and binding properties (17). All operons encoding C/U fimbriae consist at least of a fimbrial main subunit, a specific periplasmic chaperone, and an usher located in the outer membrane (18). Curli fimbriae are encoded by two operons, *csgBAC* and *csgDEFG*, and are assembled by the nucleation-precipitation pathway. Curli fimbriae are known to be involved in the formation of biofilms together with cellulose (19, 20). T1SS-secreted SiiE mediates the first contact of *S.* Typhimurium to polarized epithelial cells, which is followed by invasion mediated by the SPI1-encoded T3SS and its effector proteins (21, 22). T1SS-secreted BapA is involved in biofilm formation (20, 23). PagN and Rck were reported to mediate SPI1-T3SS-independent, zipper-like invasion of mammalian cells (24, 25). Adhesive properties of these OMPs were proposed, which might also affect adhesion to nonmammalian organisms. Monomeric autotransporters MisL and ShdA and trimeric autotransporter SadA are not expressed under laboratory conditions, and little is known about their native expression (26). MisL and ShdA are known to bind fibronectin and to be involved in intestinal infection of mice (27–29). In addition to adhesins proper as possible factors involved in adhesion to *L. sativa* leaves lipopolysaccharide (LPS), flagella filament, and motility were considered.

For most adhesins, general environmental or host factors inducing expression are not known. Therefore, we used the synthetic Tet-on system based on the *tetA* promoter for controlled expression of various *S.* Typhimurium adhesins (26, 30). We tested the adhesive structures for involvement in binding to *L. sativa* leaf surfaces by a reductionist approach described before (31), enabling direct comparison between *S.* Typhimurium interaction with *V. locusta* or *L. sativa*. Our data will support the classification of adhesive structures commonly involved in binding to leafy green salad species in order to subsequently develop strategies for prevention or reduction of adhesion of *S. enterica* to leaves of leafy green salads.

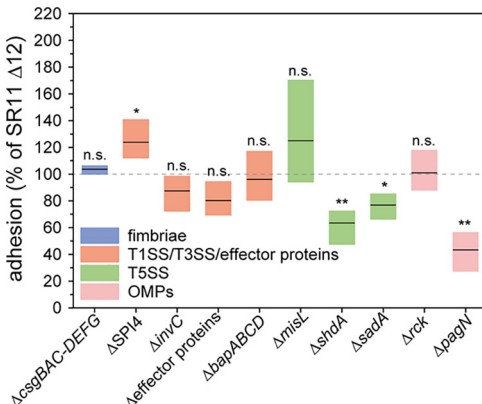

**FIG 1** Impact of deletions of genes encoding putative adhesive structures and effector proteins of SPI1-T3SS on adhesion of *S.* Typhimurium to lettuce. *L. sativa* was grown under aseptic conditions, and leaf discs were generated and inoculated with *S.* Typhimurium SR11 Δ12 or various SR11 Δ12 strains with additional deletions in genes encoding putative adhesive structures and effector proteins of the SPI1-T3SS (Δ*sopA* Δ*sopB* Δ*sopD* Δ*sopE2* Δ*sipA* [Δeffector proteins]). For the quantification of adherent bacteria, leaf discs were homogenized in PBS containing 1% deoxycholate, and serial dilutions of the homogenate and inoculum were plated onto MH agar plates for the quantification of colony-forming units (CFU). Levels of adhesion were determined as the percentage of inoculum CFU recovered in leaf disc homogenates, and adhesion of various strains was normalized by SR11 Δ12 set to 100% adhesion. The results of three biological replicates are represented as box plots, with means shown as black lines and boxes indicating upper and lower quartiles. Statistical significances between adhesion of *S.* Typhimurium SR11 Δ12 and *S.* Typhimurium SR11 Δ12 with additional deletion as indicated were calculated by Student's *t* test and are indicated as follows: ns, not significant; *, $P < 0.05$; **, $P < 0.01$; and ***, $P < 0.001$.

## RESULTS

**A synthetic system for analyses of *Salmonella enterica* adhesion to *L. sativa* leaves.** In a previous study (31), we deployed a reductionist, synthetic approach to identify factors that contribute to the surface binding of *S.* Typhimurium to the leaves of *V. locusta*. We now investigate factors that contribute to surface binding of *S.* Typhimurium to leaves of lettuce. Inoculation of *V. locusta* and *L. sativa* grown under aseptic conditions with *S.* Typhimurium indeed revealed higher levels of adhesion of *S.* Typhimurium to *L. sativa* (Fig. S1A). To identify adhesive structures required for adhesion to multiple or individual leafy green salad species, we investigated the complex adhesome of *S.* Typhimurium in its entirety.

Various databases such as GEO, SalCom, and others (12, 32, 33) were used to identify potential environmental stimuli that induce expression of *S.* Typhimurium adhesins (data not shown). However, these analyses only revealed that 3 out of 20 adhesins in the *S.* Typhimurium adhesome (i.e., Saf, Sii, PagN) defined culture conditions leading to expression. The different inducing conditions excluded a systemic comparison. Therefore, we expressed various adhesins ectopically under the control of the *tetR* P*tetA* cassette as previously described (26). To avoid interference with native expression of certain adhesins, we decided to use strain SR11 Δ12 lacking all 12 C/U fimbriae. *S.* Typhimurium SR11 Δ12 strains harboring plasmids for expression of individual adhesins were analyzed for levels of adhesion to lettuce. Further, we tested strains lacking additional putative adhesive structures such as flagella, O-antigen (OAg) of LPS, and a strain lacking all 20 known adhesive structures (Δ12 Δ*misL* Δ*sadA* Δ*shdA* ΔSPI4 Δ*bapABCD* Δ*rck* Δ*pagN* Δ*csgBAC-DEFG* [SR11 Δ20]). Adhesion assays with *L. sativa* leaves revealed no differences in adhesion for SR11 WT and the SR11 Δ12 lacking all 12 C/U fimbriae (Fig. S1B). Furthermore, the deletion of all putative adhesive structures in strain SR11 Δ20 did not alter adhesion compared to SR11 Δ12 (Fig. S1C). For all further experiments analyzing *S.* Typhimurium adhesion to *L. sativa* leaves, SR11 Δ12 was used as control. Adhesion of strains harboring further gene deletions (Fig. 1) or harboring plasmids for anhydrotetracycline (AHT)-induced expression (Fig. 2 to 6) of putative adhesive structures was normalized to 100% SR11 Δ12 to be able to compare between various experiments since sample preparation and handling of *L. sativa* leaf discs did not allow testing of

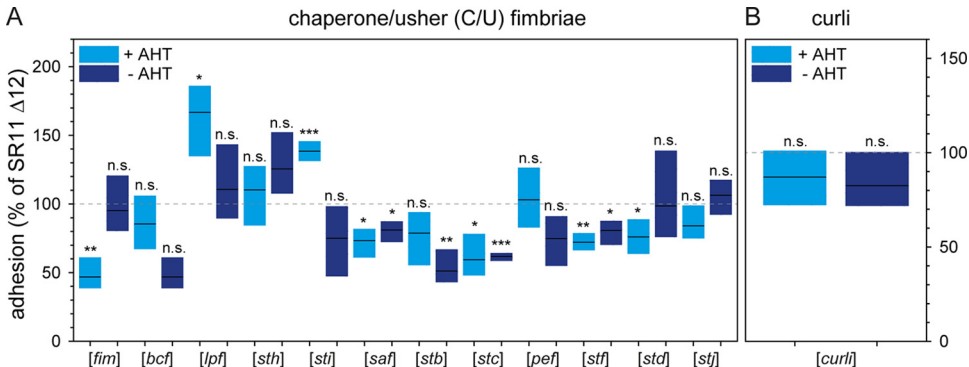

**FIG 2** Impact of expression of chaperone-usher fimbriae or curli fimbriae expression on adhesion of *S.* Typhimurium to lettuce. Sterile grown *L. sativa* was inoculated with *S.* Typhimurium SR11 Δ12 or *S.* Typhimurium SR11 Δ12 strains harboring plasmids for the expression of various chaperone-usher fimbriae (A) or curli fimbriae (B). If indicated, expression of fimbriae was induced by addition of 10 ng/mL AHT during subculture for 3.5 h. Adhesion was determined as described in Fig. 1, and the results of three biological replicates are represented as box plots, with means shown as black lines and boxes indicating upper and lower quartiles. Statistical significances between adhesion of *S.* Typhimurium SR11 Δ12 and *S.* Typhimurium SR11 Δ12 harboring plasmids as indicated with or without AHT induction were calculated by Student's *t* test and are indicated as follows: ns, not significant; *, $P < 0.05$; **, $P < 0.01$; and ***, $P < 0.001$.

more than 11 samples (3 leaf discs per sample) in one experiment. A comparison between AHT-induced and noninduced samples of a certain strain was included in one experiment.

**Contribution of fimbrial adhesins to adhesion to lettuce.** We analyzed adhesion to *L. sativa* by AHT-induced expression of all C/U fimbriae encoded by *S.* Typhimurium, one at a time. Different levels of binding to *L. sativa* leaves were mediated by expression of C/U fimbriae (Fig. 2A). Expression of Bcf, Sth, Stb, Pef, and Stj did not affect adhesion compared to background strain SR11 Δ12. Therefore, cognate ligands may be absent on *L. sativa* leaves. The expression of Fim, Saf, Stc, Stf, and Std fimbriae led to decreased adhesion (47%, 73%, 59%, 72%, and 76% means, respectively). However, SR11 Δ12 harboring plasmids encoding Saf, Stc, and Stf fimbriae also resulted in decreased adhesion in the absence of inducer AHT. In the absence of AHT, no synthesis and surface expression of these three fimbriae were observed by flow cytometry (26). Expression of Lpf and Sti fimbriae resulted in significantly increased adhesion to *L. sativa* (means of 167% and 139%, respectively). Noninduced controls for Lpf and Sti fimbriae showed similar adhesion to background SR11 Δ12. AHT-induced expression of curli fimbriae did not alter adhesion (Fig. 2B), as well as the deletion of *csgBAC-csgDEFG* did not alter adhesion to *L. sativa* leaves compared to SR11 Δ12 (Fig. 1).

**Contribution of T1SS-secreted nonfimbrial adhesins to adhesion to lettuce.** A small subpopulation of *S.* Typhimurium expresses SiiE under laboratory conditions (3.5 h subculture in LB). Enhanced surface expression of SiiE was achieved by AHT-induced expression of *hilD*, the central transcriptional activator of SPI1/SPI4 genes (34), as generation of a vector for Tet-on expression of the *sii* operon failed. The frequency of SiiE-expressing *S.* Typhimurium increased from approximately 12% under native conditions to over 80% after AHT-induced *hilD* expression (31). The increased surface expression of SiiE in SR11 Δ12 correlated with significantly decreased adhesion to *L. sativa* (Fig. 3A), while *S.* Typhimurium ΔSPI4 showed significantly increased adhesion (Fig. 1). Since *hilD* expression also affects the expression of SPI1 and associated effector proteins, we tested the expression of *hilD* in strains harboring deletions in SPI4 or *invC* encoding the ATPase subunit of SPI1-T3SS. In both cases, increased expression neither of SPI1-T3SS genes in *S.* Typhimurium ΔSPI4 nor SPI4 genes in *S.* Typhimurium Δ*invC* led to altered adhesion compared to background strain SR11 Δ12. Therefore, the increased simultaneous expression of genes in SPI4 and SPI1 possibly impaired adhesion to *L. sativa* leaves.

AHT-induced expression of *bapABCD* led to significantly increased adhesion to lettuce (Fig. 3B), whereas deletion of *bapABCD* did not affect adhesion (Fig. 1). To achieve

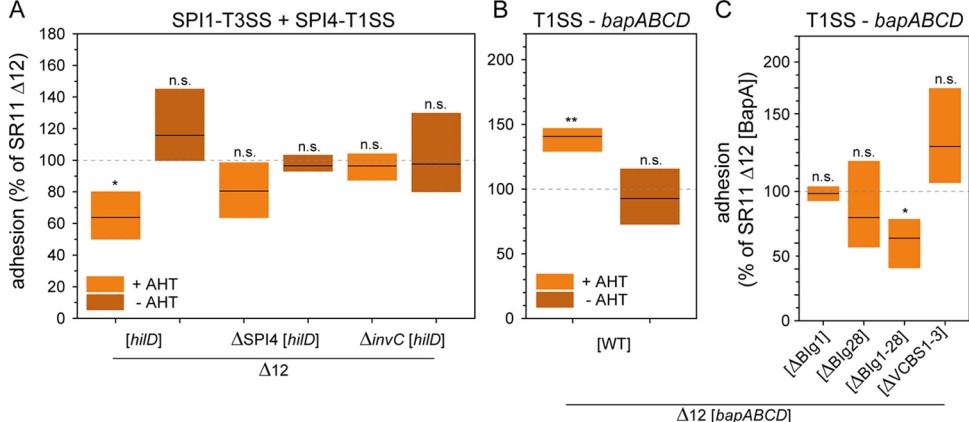

**FIG 3** Impact of expression of T1SS-secreted adhesins or *hilD* on adhesion of *S.* Typhimurium to lettuce. (A) *L. sativa* grown under aseptic conditions was inoculated with *S.* Typhimurium strains SR11 Δ12, SR11 Δ12 ΔSPI4, and SR11 Δ12 Δ*invC*, each with overexpression of regulator *hilD* for analysis of the SPI4-encoded T1SS-secreted adhesin SiiE and the SPI1-encoded T3SS. (B and C) Adhesion of SR11 Δ12 stains with surface expression of T1SS-secreted adhesin BapA (WT BapA, full length) (B) or truncated forms of BapA (C). Expression of adhesins was induced by addition of AHT if indicated. The mean levels of adhesion and statistical significances were determined as described in Fig. 2.

a better understanding of BapA binding to *L. sativa* leaves, we used various truncated forms of BapA (31). Alleles of *bapA* with deletions of BIg1, BIg28, or BIg1-28 and VCBS1-3 were expressed by AHT induction. Synthesis and secretion of mutant forms of BapA were confirmed by flow cytometry in prior work (31) for ΔBIg1, ΔBIg28, or ΔBIg1-28 and revealed that deletion of BIg1-28 ablated surface expression of BapA. Synthesis and secretion of the mutant form of BapA lacking VCBS domains 1 to 3 were confirmed by flow cytometry in this study (Fig. S2). The deletion of BIg1 and BIg28 showed no significantly altered adhesion compared to WT-BapA, which could also be observed for the deletion of all VCBS domains (Fig. 3C). Deletion of all 28 BIg domains resulted in significantly decreased adhesion, which is in line with the loss of BapA surface expression.

**Contribution of OMP adhesins to adhesion to lettuce.** AHT-induced expression of Rck, as well as deletion of *rck*, resulted in no altered adhesion to *L. sativa* leaves (Fig. 1 and Fig. 4A). Expression of *pagN* led to a significantly reduced adhesion (52% mean), which was also observed for the noninduced samples (67% mean). However, Western blot analyses confirmed the absence of PagN in noninduced cultures (26). In addition, a strain defective in *pagN* revealed a significantly reduced adhesion (Fig. 1, 43% mean).

**Contribution of autotransported adhesins to adhesion to lettuce.** The AHT-induced expression of *shdA* led to a significantly decreased adhesion to *L. sativa* leaves

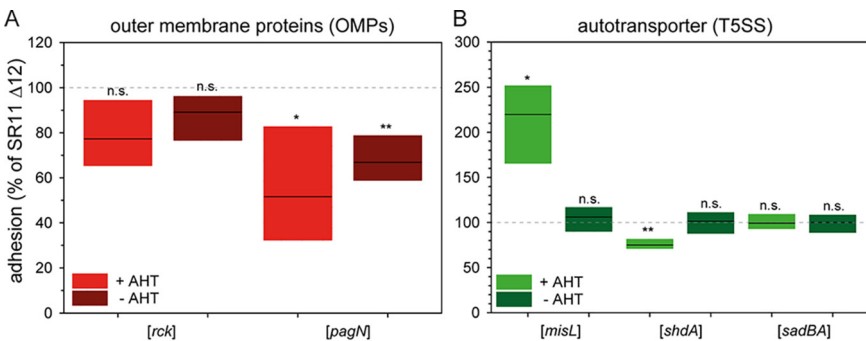

**FIG 4** Impact of expression of outer membrane proteins or T5SS-secreted adhesins on adhesion of *S.* Typhimurium to lettuce. (A) For the analysis of outer membrane proteins, SR11 Δ12 strains expressing *rck* or *pagN* were used. Expression of the adhesins was induced by addition of AHT if indicated, as described in Fig. 1. (B) *L. sativa* grown under aseptic conditions was inoculated with *S.* Typhimurium strain SR11 Δ12 or *S.* Typhimurium strain SR11 Δ12 with surface expression of T5SS-secreted adhesins MisL, ShdA or SadA. The levels of adhesion and statistical significance were determined as described in Fig. 2.

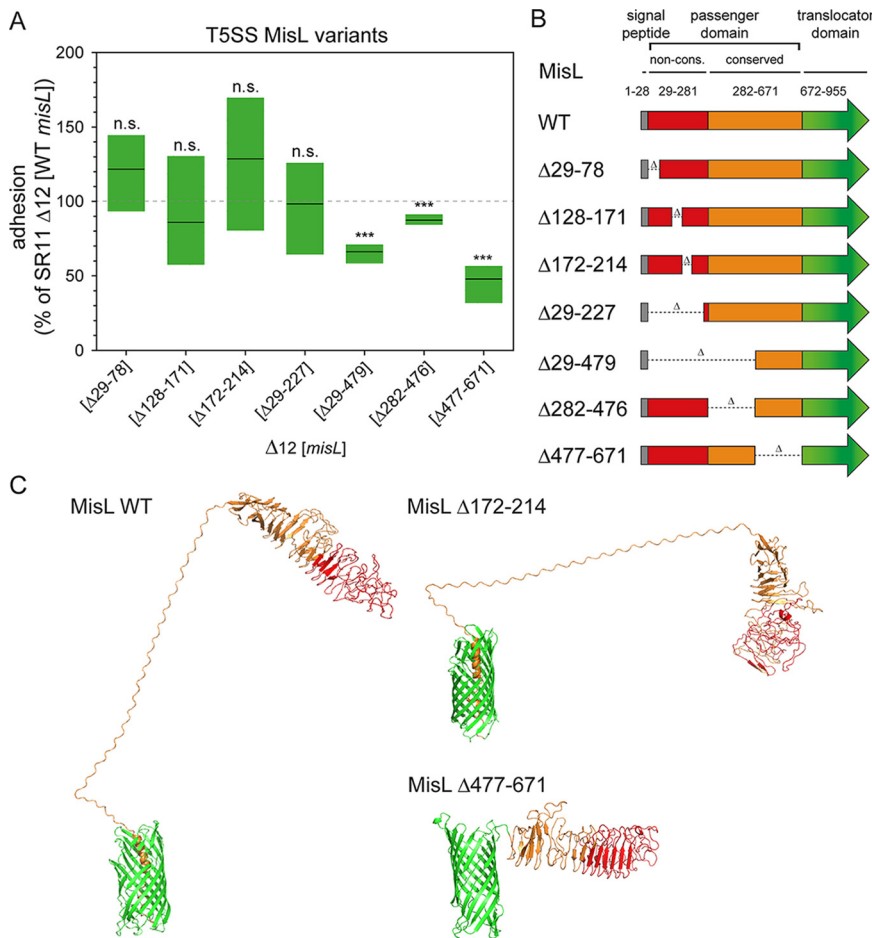

**FIG 5** Impact of expression of MisL on adhesion of *S.* Typhimurium to *L. sativa* and mutational analysis of MisL. (A) *L. sativa* grown under aseptic conditions was inoculated with *S.* Typhimurium strain SR11 Δ12 with surface expression of T5SS-secreted adhesin MisL or truncated forms of MisL. Expression of the adhesins was induced by addition of AHT. Adhesion was normalized by SR11 Δ12 (*misL* WT) set to 100% adhesion. Statistical significance was determined as described in Fig. 2. (B) Models for domain organization of WT MisL and various mutated forms used in adhesion assays. (C) Homology-based protein structure prediction by trRosetta for MisL WT and mutated forms of MisL. Visualization by PyMol. Colors represent translocator domain (green), conserved passenger domain (orange), and nonconserved passenger domain (red).

(75% mean) (Fig. 4B), as well as deletion of *shdA* (63% mean) (Fig. 1). AHT-induced expression of *sadBA* did not alter adhesion to lettuce, but deletion of *sadA* resulted in significantly reduced adhesion (Fig. 1, 77% mean). The AHT-induced expression of *misL* revealed a significant increased adhesion to *L. sativa* compared to background strain SR11 Δ12 (Fig. 4B).

To gain further insight into the contribution of MisL in binding to *L. sativa* leaves, we decided to test truncated forms of MisL in adhesion to *L. sativa* (Fig. 5). MisL contains a signal sequence for Sec-dependent secretion into the periplasm (amino acids [aa] 1 to 28) and a translocation domain (aa 677 to 955) forming a β-barrel in the outer membrane and mediating transport of the passenger domain (aa 29 to 676) across the outer membrane. The passenger domain is further divided into a nonconserved region and a conserved region compared to other, nonprotease autotransported adhesins (27). Dorsey et al. (27) revealed that the nonconserved passenger domain (aa 29 to 281) is responsible for binding of MisL to fibronectin and with lower affinity to collagen IV. Therefore, we created mutant forms of MisL with deletions in the nonconserved passenger domain (aa 29 to 281) in the conserved passenger domain (aa 282 to 671), and a form lacking the entire nonconserved and partly the conserved passenger

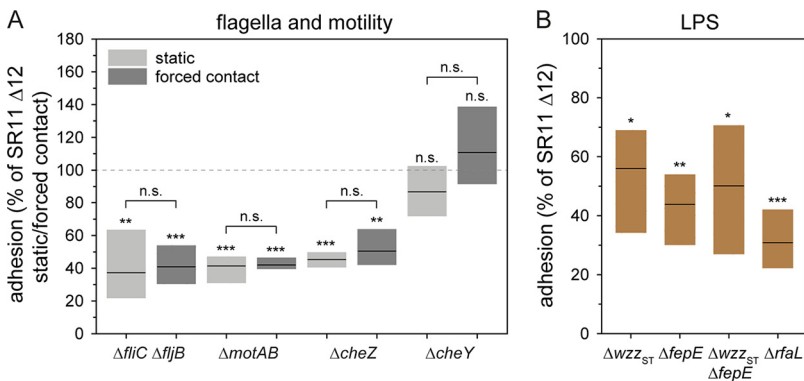

**FIG 6** Impact of defects in motility or flagella assembly or alteration in O-antigen length on *S.* Typhimurium adhesion to lettuce. (A) *L. sativa* grown under aseptic conditions was inoculated with *S.* Typhimurium SR11 Δ12, or *S.* Typhimurium SR11 Δ12 with deletion of various genes required for motility or chemotaxis. (B) Effect of mutations in genes resulting in lack (*rfaL*) or various truncations of O-antigen length (*wzz*$_{ST}$, *fepE*, *wzz*$_{ST}$ *fepE*). Inoculation was performed under static conditions (A, B) or by forced contact by centrifugation at 500 × *g* for 5 min to compensate effects of mutations in motility genes (A). The adhesion and the statistical significance were determined as described in Fig. 2. Brackets in panel A indicate statistical significance between infection under static conditions versus forced contact.

domain (Fig. 5B). Protein structure prediction of MisL WT and truncated forms of MisL were done using trRosetta and visualized by PyMol (Fig. 5C and Fig. S4). Synthesis and secretion of truncated forms of MisL were quantified by flow cytometry, confirming the presence on the bacterial surface (Fig. S3A and B). Furthermore, potential effects on bacterial autoaggregation were investigated by microscopy (Fig. S3C) and revealed a lack of autoaggregation upon AHT induction. Deletion of the nonconserved transporter domain (Δ29-78, Δ128-171, Δ172-214, and Δ29-227) led to adhesion to *L. sativa* leaves comparable to WT MisL (Fig. 5). Deletion of large portion of the conserved passenger domain (Δ282-476, Δ477-671) led to a significantly decreased adhesion to *L. sativa* leaves compared to WT MisL. The same results were obtained for MisL Δ29-479.

Besides induced synthesis of MisL, deletion of *misL* also led to nonsignificant increased adhesion to *L. sativa* compared to background strain SR11 Δ12 (Fig. 1). However, *S.* Typhimurium Δ*misL* did not reach adhesion as observed after induced expression of *misL*.

**Contribution of flagella filaments and motility to adhesion to lettuce.** Previous studies revealed the contribution of flagella filaments and motility of *S.* Typhimurium in adhesion to various plant species (13, 35, 36). Here, we investigate the binding properties of the flagella filament and the contribution of motility in the adhesion to *L. sativa* leaves by using four distinct deletion strains. Loss of the flagella filaments (Δ*fliC* Δ*fljB*) resulted in significantly decreased adhesion to *L. sativa* leaves (37% mean) (Fig. 6A). To bypass reduced interaction with leaf surfaces possibly resulting from loss of motility, contact was forced by centrifugation (5 min at 500 × *g*). However, centrifugation did not restore adhesion of *S.* Typhimurium Δ*fliC* Δ*fljB* to leaf surfaces (41% mean). To analyze the contribution of flagella filaments as putative adhesive structure present on the bacterial surface as shown for *Escherichia coli* flagella (35), we used SR11 Δ12 Δ*motAB* still harboring flagella filaments, but defective in flagella motor energization. This strain revealed significantly reduced adhesion in static and centrifuged samples (41% and 42% means, respectively). Therefore, the presence of flagella filaments without rotation is insufficient for adhesion to lettuce, thus not contributing to adhesion to lettuce. Deletion of *cheZ* resulting in motility restricted to tumbling also exhibited a significantly decreased adhesion to *L. sativa* (mean 45% and 50% for static and centrifuged samples, respectively). A strain defective in *cheY*, restricted to smooth swimming, showed a mean adhesion to *L. sativa* leaves under static conditions less

than 100% (87% mean), but the difference was not significant. After centrifugation, no altered adhesion (111% mean) was detected for Δ*cheY* compared to parental strain SR11 Δ12. Thus, we conclude a contribution of chemotactic motility of *S.* Typhimurium for adhesion to *L. sativa* leaves.

**Contribution of O-antigen to adhesion to lettuce.** A main constituent of the Gram-negative cell envelope is LPS, which stabilizes the cell envelope and protects bacteria against various environmental factors. Furthermore, LPS increases the negative charge of the cell envelope, and a putative adhesive role was reported (37). To analyze the contribution of LPS in adhesion to *L. sativa* leaves, we used mutant strains lacking various genes involved in the biosynthesis and control of the length of modal repeats of the OAg of LPS (Fig. 6B). In *S.* Typhimurium WT, a heterogeneous distribution of the short-chain OAg (S-OAg), long-chain OAg (L-OAg), and very long-chain OAg (VL-OAg), was found. Deletion of $wzz_{ST}$ results in the homogenous distribution of only VL-OAg, whereas in *S.* Typhimurium Δ*fepE*, a homogenous distribution of only L-OAg is present. A strain defective in both genes (Δ$wzz_{ST}$ Δ*fepE*) only possesses S-OAg. Deletion of *rfaL* results in LPS with a core oligosaccharide (OS) lacking OAg. All deletion strains showed a significantly reduced adhesion to *L. sativa* leaves compared to background strain SR11 Δ12 (Fig. 6B, i.e., means of 56% for Δ$wzz_{ST}$, 44% for Δ*fepE*, 31% for Δ$wzz_{ST}$ Δ*fepE*, and 50% for Δ*rfaL*). Hence, the heterogeneous distribution of L-OAg and VL-OAg on the bacterial surface is an important factor in the adhesion to *L. sativa* leaves.

## DISCUSSION

Here, we addressed which factors of *S. enterica* are involved in adhesion to plant surfaces by using a reductionist, synthetic approach with controlled surface expression of specific adhesive structures. All known adhesive structures encoded by *S.* Typhimurium were tested for their impact on adhesion to *L. sativa* leaves. The results of this study are summarized in Fig. 7.

The involvement of flagella of *S.* Typhimurium in adhesion to various leafy green salad species has been investigated before in several studies that reported decreased adhesion to basil, lettuce, and *V. locusta* leaves for mutant strains lacking the flagella filament (13, 31, 38). These results are partly in line with our findings for decreased adhesion to *L. sativa* leaves of an *S.* Typhimurium mutant strain lacking flagella filaments. Rossez et al. (35) investigated the binding of pathogenic and nonpathogenic *E. coli* strains on a molecular level, revealing ionic binding of sulfated and phosphorylated plasma membrane lipids on *Arabidopsis thaliana* leaves. Here, we revealed an involvement of flagella-mediated motility in adhesion of *S.* Typhimurium to *L. sativa*, but not by the flagella itself. However, whether *S.* Typhimurium flagella filaments also bind sulfated and phosphorylated plasma membrane lipids of *Arabidopsis thaliana* leaves still has to be investigated. In addition to the flagella filament, Kroupitski et al. (13) found a contribution of chemotaxis since a Δ*cheY* strain ("run only") showed decreased internalization in *L. sativa* leaves. These authors hypothesized that lack of chemotaxis for higher sucrose concentrations close to stomata is affected, whereas the initial attachment of *L. sativa* leaves was not affected. We investigated the involvement of motility in adhesion to *L. sativa* leaves. A decreased adhesion was shown for strains deficient in energization (Δ*motAB*) or switching from clockwise (CW) to counterclockwise (CCW) flagella rotation (Δ*cheZ*, "tumbling only"), thus indicating the importance of directed motility. We hypothesize that with lack of directed motility, *S.* Typhimurium fails to reach preferred areas on leaf surfaces of *L. sativa*. Even after forced contact, adhesion of *S.* Typhimurium Δ*cheZ* was not restored to WT levels, indicating the importance of CCW rotation for smooth swimming to reach preferred binding sites. In contrast to Δ*cheZ*, Δ*cheY* (CCW, only run) is not significantly affected in adhesion to *L. sativa* leaves, as shown by Kroupitsli et al. (13), probably by reaching preferred binding sites on *L. sativa* leaves due to constantly "swimming and searching." Adhesion for Δ*cheY* could also be enhanced by centrifugation, forcing contact of bacteria with the leaf surface, in contrast to more random contacts by bacteria swimming in medium. A role of

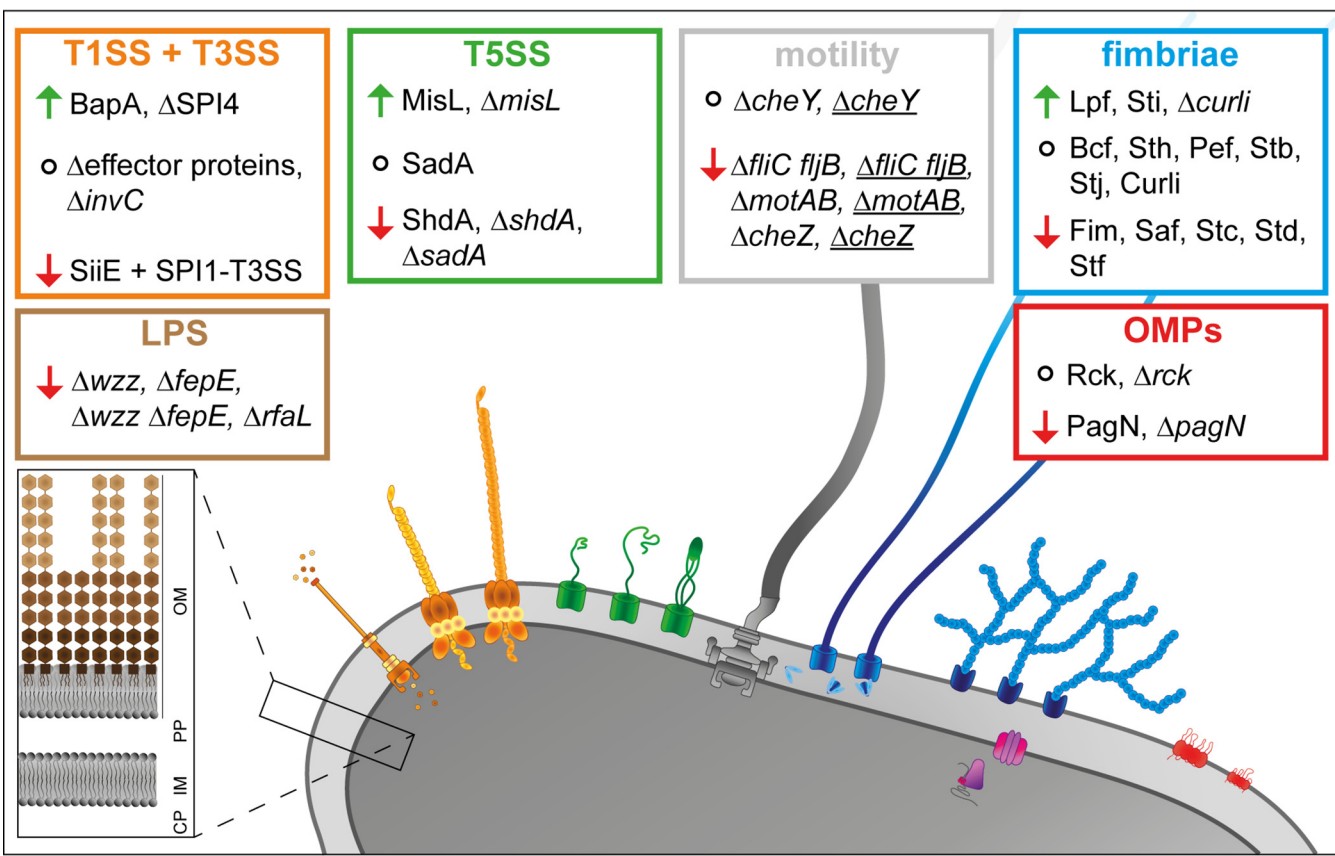

**FIG 7** Overview of the role of analyzed factors in the adhesion of *S.* Typhimurium to *L. sativa* leaves. Nonunderlined and underlined acronyms indicate static or centrifuged samples (forced contact), respectively. Arrows indicate increased (green) or decreased (red) adhesion, and circles indicate that adhesion was not altered. OM, outer membrane; PP, periplasm; IM, inner membrane; CP, cytoplasm.

chemotactic control in *S.* Typhimurium for interaction with mammalian epithelial was observed and considered to be mediated by altered frequency of contacts (39). In summary, *S.* Typhimurium adhesion to *L. sativa* leaves depends on motility and, in particular, on the ability to switch flagella rotation from CW (tumble) to CCW (run).

The importance of LPS for *S.* Typhimurium interaction with mammalian cells was investigated in several prior studies (40, 41). Here, we elucidated the importance of LPS with full-length O-antigen for *S.* Typhimurium adhesion to *L. sativa* leaves and observed that LPS structure is an important factor. The data indicate that LPS exhibits either specific binding properties for *L. sativa* leaves, or, more likely, the surface charge and/or hydrophilicity of *S.* Typhimurium with intact LPS mediates adhesion to *L. sativa* leaves. This explanation for our finding of *S.* Typhimurium LPS involvement in adhesion to *V. locusta* leaves is in line with a previous study (31). In studies of adhesion of pathogenic *E. coli*, lower attachment of LPS mutants to *A. thaliana* and romaine lettuce leaves was observed (42). Further, De Moraes et al. (43) showed the involvement of *S.* Typhimurium LPS also for the persistence in tomatoes, revealing an impact on the entire infection process. However, changes of the bacterial cell surface charge due to altered LPS structure were not shown yet, but altered biofilm formation was observed for plant-pathogenic bacteria with altered LPS structure. For example, the altered LPS structure of *P. aeruginosa* caused decreased biofilm formation and affected virulence (44). Nevertheless, the involvement of LPS of *S.* Typhimurium in adhesion to plants and effects of altered O-antigen structure on surface properties have to be further investigated.

In addition, the expression of fimbrial adhesins revealed increased adhesion to *L. sativa* leaves if Sti or Lpf fimbriae were expressed; hence, we propose cognate ligands on *L. sativa* leaves for Lpf and Sti fimbriae. While expression of Sti or Lpf was observed during infection of bovine ligated ileal loops (17), no specific ligands are known for

these fimbriae so far. Whereas Sti fimbriae are absent in *S. enterica* serovar Typhi and some strains of *S. enterica* serovar Paratyphi A, Lpf fimbriae are restricted to *S.* Typhimurium (45). Bäumler et al. (46) showed the involvement of Lpf fimbriae in adhesion to murine Peyer's patches and HEp-2 cells (46), and Lpf fimbriae contribute to long-term intestinal colonization of *S.* Typhimurium in resistant mice (47). Additionally, Ledeboer et al. (48) revealed 10-fold-increased amounts of LpfE in biofilms, leading to the assumption that Lpf fimbriae are involved in biofilm formation during the microcolony stage. Moreover, deletion of Lpf fimbriae resulted in loss of biofilm formation on chicken intestinal epithelium. Little is known about Sti fimbriae, though a study by Laniewski et al. (49) revealed the involvement of Sti, Saf, Stc, and curli fimbriae in a murine infection model. Deletion of all four fimbrial operons resulted in a strain attenuated in a murine infection model, but if only one operon was deleted, no effect was observed. This indicates a possible redundancy of these fimbriae in murine infection, which were not observed in our synthetic model of adhesion to *L. sativa* leaves. Binding properties of Sti and Lpf fimbriae have to be further investigated, i.e., by glycoarrays (50). However, expression of Fim fimbriae led to decreased adhesion to *L. sativa* leaves as observed before for adhesion to *V. locusta* (31). We hypothesize that the highly abundant surface of Fim fimbriae (compare reference 26) may mask other structures on the bacterial surface involved in adhesion to *L. sativa* leaf surfaces.

Deletion of SPI4 encoding SiiE and its cognate T1SS led to increased adhesion, whereas overexpression of SPI4 genes by overexpression of regulator *hilD* did not alter adhesion compared to the background strain. The same results were observed for adhesion to *V. locusta* leaves (31). In addition, native expression of the SPI4/SPI1 regulon is tightly controlled by host cell factors (21), and the selective control of expression also argues against an involvement in adhesion to *L. sativa* leaf surfaces.

We showed involvement of BapA in adhesion to *L. sativa* leaves. Whereas binding properties for specific glycostructures of BapA remain to be determined, several studies revealed a contribution of BapA in biofilm formation (20, 23). In this study, plant surfaces were inoculated for 1 h; thus, effects of BapA are unlikely due to biofilm formation that requires longer incubation. In our previous study, we revealed the involvement of BapA expression in adhesion to *V. locusta* leaves leading to increased adhesion compared to background strain (31). Further, we tested the involvement of various BIg domains by using mutant alleles of BapA and revealed that adhesion of BapA to *V. locusta* leaves was reduced by deletion of at least one BIg domain, indicating that the length of this adhesin is critical. Mutant alleles of BapA lacking BIg1 or BIg28 were not altered in adhesion to *L. sativa* leaves. Deletion of all 28 BIg domains led to a decreased adhesion compared to WT BapA, suggesting length dependency of adhesion to *L. sativa* leaves since allele BapA ΔBIg1-28 only consisting of N-terminal and C-terminal portions, including the VCBS domains 1 to 3, might not project a putative binding domain beyond the O-antigen layer of LPS. However, BIg domains other than BIg1 and BIg28 might be involved in binding to *L. sativa* leaves and are absent in BapA ΔBIg1-28. Deletion of the VCBS domains 1 to 3 did not show a significant difference in adhesion compared to WT BapA; thus, the putative adhesive VCBS domains might not be involved in binding to *L. sativa* leaves. In comparison to other pathogens harboring adhesins with comparable VCBS domains, these domains might be involved in biofilm formation and adhesion to roots with subsequent biofilm formation (51, 52). However, these adhesins encoded by *Vibrio fischeri* (LapV) and *Pseudomonas putida* (LapF) possess much larger numbers of VCBS domain repeats (32 and 64, respectively). Moreover, *bapABCD* is highly conserved among *S. enterica* serovars (with some variations in *bapA*), indicating an important role for *S. enterica* lifestyle outside mammalian hosts and possibly during infection of mammalian hosts (53). Contribution of BapA encoded by other *S. enterica* serovars in adhesion to various leafy green salad species has to be tested in addition to the influence of longer inoculation times and involvement of BapA in biofilm formation on leafy green salad leaves. Here, we suggest a general role of BapA of *S.* Typhimurium in adhesion to leafy green salads.

In this study, we showed an involvement of autotransported MisL in adhesion to *L. sativa* leaves. Expression of MisL is known to facilitate adhesion to epithelial cell lines CaCo2 and HeLa to enhance biofilm formation (54) and to contribute to intestinal colonization of mice (27). Furthermore, binding specificity of the nonconserved passenger domain (aa 29 to 281) for fibronectin and, with lower affinity for collagen IV, was determined in extracellular matrix protein-binding assays (27). Since deletion of the nonconserved passenger domain of MisL did not alter adhesion to *L. sativa* leaves compared to WT MisL, we hypothesize distinct binding properties for epithelial cells and *L. sativa* leaf surfaces. We suggest that impaired binding mediated by MisL with deletions in the conserved passenger domain is due to truncation of the long stalk, resulting in inability to interact with cognate binding partners on the *L. sativa* leaf surface (Fig. 5C, compare structure of truncated MisL with deletion of aa 477 to 671). However, deletions of aa 282 to 476 or aa 29 to 80 led to MisL alleles that still harbored a long stalk but might lack a binding domain (Fig. 5C, compare structure of truncated MisL with deletion of aa 282 to 476 and aa 29 to 479). In conclusion, the nonconserved domain of MisL involved in binding to mammalian fibronectin and collagen IV is not involved in binding to *L. sativa* leaves. Therefore, the ligands for MisL on *L. sativa* leaves remain to be identified. The relevance of MisL in colonization of *L. sativa* was previously investigated by Kroupitski et al. (14). A genetic screen identified, together with *stfC* (encoding C/U fimbriae subunit) and *bcsA* (cellulose synthase subunit), *misL* as upregulated in *S.* Typhimurium on *L. sativa* after storage for 7 days at 8°C. Deletion of *misL* led to decreased survival on *L. sativa* leaves under cold-storage conditions. Interestingly, the effect of *misL* on survival was dependent on the presence of *L. sativa* and absent after sole cold storage (14).

In this study, we showed, for the first time, the contribution of directed motility, an intact LPS layer, and expression of various adhesive structures of *S.* Typhimurium to adhesion to *L. sativa* leaves. We revealed that synthetic expression of Lpf or Sti fimbriae, T1SS-secreted BapA, or autotransported MisL led to enhanced adhesion to *L. sativa* leaves. To gain further insight into the adhesion of *S.* Typhimurium to leafy green salads, BapA has to be further investigated, possibly revealing common adhesive interactions with plant surfaces. Furthermore, expression of all adhesive structures, especially adhesive structures involved in adhesion to leafy green salads, have to be further examined with regard to their native expression. For this purpose, we suggest transcriptomics or proteomic analyses of *S.* Typhimurium grown under various environmental conditions. With this knowledge, culturing conditions could be adjusted to choose unfavorable conditions for *S.* Typhimurium and the expression of adhesive structures involved in the adhesion to leafy green salads. Another aspect are the binding specificities of the adhesive structures being involved in adhesion to leafy green salad leaves facilitating the initial attachment of *S.* Typhimurium to the plant. For most adhesive structures of *S.* Typhimurium, binding specificities are not known. Screens in glycan arrays with defined oligosaccharides, or with cell wall extracts from various leafy green salads, could reveal binding specificities. This was done before for fimbrial adhesins of *E. coli* (55, 56). Binding to specific ligands could be prevented by adding specific sugars (e.g., mannose for type-1-fimbriae [57]) to washing water or by choosing plant species with smaller amounts or lack of specific ligands present in the leaves.

## MATERIALS AND METHODS

**Bacterial strains and culture conditions.** Bacterial strains used in this study are listed in Table 1. Unless otherwise stated, bacteria were grown at 37°C in lysogeny broth (LB) medium or on LB agar containing antibiotics for selection of specific markers if required to maintain plasmids listed in Table 2. Carbenicillin (Carb) and kanamycin (Km) were used at final concentrations of 50 $\mu$g/mL. For the induction of the Tet-on system, anhydrotetracycline (AHT) was added to final concentrations of 10 to 100 ng/mL.

**Construction of strains and plasmids.** For the construction of plasmids encoding various truncated forms of *misL* under the Tet-on system, plasmid p4403 was used as the template. The plasmid was amplified by using oligonucleotides listed in Table S1 in the supplemental material, resulting in deletion of various amounts of codons. Amplified products were reassembled by site-directed mutagenesis (SDM) kit according to manufacturer's protocol (NEB). For the construction of the template vector p4251, vector pWSK29 and the Tet-on system consisting of *tetR* P$_{tetA}$ present on p3773 were amplified using oligonucleotides listed in Table S1.

**TABLE 1** Bacterial strains used in this study

| Designation | Relevant characteristic(s) | Source or reference |
|---|---|---|
| *E. coli* NEB5α | Cloning strain | New England Biolabs |
| *S.* Typhimurium NCTC | *S.* Typhimurium wild type | NCTC 12023, lab stock |
| *S.* Typhimurium SR11 | Wild type | 61 |
| SPN376 (referred to as SR11 Δ12) | Δ*fimAICDHF* Δ*stbABCD* Δ*sthABCDE* Δ*stfACDEFG* Δ*stiABCH* Δ*bcfABCDEFGH* Δ*safABCD* Δ*pefACDorf5orf6* Δ*stcABCD* Δ*stjEDCBA* Δ*stdAB* Δ*lpfABCDE*::KSac | 31 |
| MvP2447 | Δ12 Δ*misL*::FRT | 31 |
| MvP2506 | Δ12 Δ*rck*::*aph* I-SceI | 31 |
| MvP2507 | Δ12 Δ*pagN*::*aph* I-SceI | 31 |
| MvP2622 | Δ12 Δ*shdA*::*aph* | 31 |
| MvP2623 | Δ12 Δ*sadA*::*aph* | 31 |
| MvP2624 | Δ12 ΔSPI4::*aph* | 31 |
| MvP2625 | Δ12 Δ*bapABCD*::*aph* | 31 |
| MvP2703 | Δ12 Δ*csgBAC-DEFG*::*aph* | 31 |
| MvP2707 (referred to as Δ20) | Δ12 Δ*misL* Δ*sadA* Δ*shdA* ΔSPI4 Δ*bapABCD* Δ*rck* Δ*pagN* Δ*csgBAC-DEFG* | 31 |
| MvP2710 | Δ12 Δ*misL* Δ*sadA* Δ*shdA* ΔSPI4 Δ*bapABCD* Δ*rck* Δ*pagN* Δ*csgBAC-DEFG* Δ*fliI*::*aph* | 31 |
| MvP2711 | Δ12 Δ*misL* Δ*sadA* Δ*shdA* ΔSPI4 Δ*bapABCD* Δ*rck* Δ*pagN* Δ*csgBAC-DEFG* Δ*motAB*::*aph* | 31 |
| MvP2718 | Δ12 Δ*invC*::*aph* | 31 |
| MvP2751 | Δ12 Δ*cheZ*::*aph* | 31 |
| MvP2754 | Δ12 Δ*cheY*::*aph* | 31 |
| MvP2757 | Δ12 Δ*motAB*::*aph* | 31 |
| MvP2763 | Δ12 Δ*fliC*::FRT Δ*fljB*::*aph* | 31 |
| MvP2798 | Δ12 Δ*fepE*::FRT | 31 |
| MvP2799 | Δ12 Δ*wzz*$_{ST}$::FRT | 31 |
| MvP2800 | Δ12 Δ*rfaL*::FRT | 31 |
| MvP2812 | Δ12 Δ*wzz*$_{ST}$::*aph* Δ*fepE*::FRT | 31 |
| MvP2844 (=SR11 Δ effector proteins) | Δ12 Δ*sopB*::FRT Δ*sopA*::FRT Δ*sopE2*::FRT Δ*sopD*::FRT Δ*sipA*::*aph* | 31 |

The PCR products were purified and assembled by Gibson assembly (GA) according to the manufacturer's protocol (NEB). For the construction of the truncated form of *bapA* lacking VCBS domains 1 to 3 under the Tet-on system, template vector p4251 was used, and *bapABCD* without VCBS domains 1 to 3 was amplified by PCR from the genome of *S.* Typhimurium NCTC 12023 using oligonucleotides listed in Table S1. Three PCR products for portions or *bapABCD* and vector p4251 were assembled by GA. Sequence-confirmed plasmids were electroporated in *S.* Typhimurium SR11 Δ12.

**Structure prediction of MisL and mutant alleles.** Structure prediction of MisL and mutant alleles was performed using the online tool trRosetta (58) with PDB template option and single-sequence input style. Predicted protein structures were then visualized using PyMol version 2.5.0 (59). TM scores ranged from 0.242 to 0.665.

**Cultivation of sterile grown salads.** *Lactuca sativa* L. cultivar seeds Tizian (butterhead lettuce) and *Valerianella locusta* seeds (Verte à cour plein 2; N.L. Chrestensen Erfurter Samen- und Pflanzenzucht) were kindly provided by Adam Schikora and Sven Jechalke (Justus-Liebig University Giessen). *L. sativa* seeds were sterilized with 3% NaClO for 4 min while continuously manually inverting the tubes (1.5-mL disposable reaction tube; Sarstedt Safelock). Further, *L. sativa* seeds were washed four times with sterile double-distilled water (ddH$_2$O) and directly planted on Murashige-Skoog (MS) agar (1.1 g MS medium, including vitamins [Duchefa Biochemie; catalog no. M0222], 10 g agar, and 5 g sucrose per liter) in sterile plastic containers with air filter (round model, 140 mm; Duchefa Biochemie; catalog no. E1674). *L. sativa* seeds were kept in the dark for 1 day at room temperature (RT) and then further cultivated at 20°C with a 12-h/12-h day-night rhythm for 4 weeks. *V. locusta* seeds were sterilized by 70% EtOH for 1 min followed by 3% NaClO for 2 min. Seeds were washed thrice with sterile ddH$_2$O and dried for 30 min. *V. locusta* seeds were planted on MS agar (2.2 g MS medium, including vitamins, 10 g agar, and 0.5 g morpholineethanesulfonic acid [MES] per L, pH 5.4) in sterile plastic containers with an air filter as above at 20°C with a 12-h/12-h day-night rhythm for 8 weeks.

**Adhesion to *L. sativa* or *V. locusta*.** Inoculation of *L. sativa* with *Salmonella* was done as previously described in detail (60), and inoculation of *V. locusta* was as described in Elpers et al. (31). Briefly, for each condition and strain, three-leaf discs of *L. sativa* or *V. locusta* were inoculated with *S.* Typhimurium strains diluted 1:31 from overnight cultures for 3.5 h (with antibiotics if required; with and without AHT for induction of Tet-on system) in test tubes with aeration in a roller drum. The cultures were diluted in phosphate-buffered saline (PBS) to obtain $5.6 \times 10^7$ bacteria/mL. Inoculation of each leaf disc with

**TABLE 2** Plasmids used in this study

| Plasmid | Relevant characteristics | Reference or source |
|---|---|---|
| p4251 | *tetR* P$_{tetA}$ in pWSK29 | 26 |
| p4253 | *tetR* P$_{tetA}$::*bapABCD* in pWSK29 | 26 |
| p4380 | *tetR* P$_{tetA}$::*csgBACEFG* in pWSK29 | 26 |
| p4389 | *tetR* P$_{tetA}$::*stiABCD* in pWSK29 | 26 |
| p4390 | *tetR* P$_{tetA}$::*stfABCDEFG* in pWSK29 | 26 |
| p4391 | *tetR* P$_{tetA}$::*stbABCDEFG* in pWSK29 | 26 |
| p4392 | *tetR* P$_{tetA}$::*fimAICDHF* in pWSK29 | 26 |
| p4393 | *tetR* P$_{tetA}$::*safABCD* in pWSK29 | 26 |
| p4394 | *tetR* P$_{tetA}$::*stdABCD* in pWSK29 | 26 |
| p4395 | *tetR* P$_{tetA}$::*stjABCDE* in pWSK29 | 26 |
| p4396 | *tetR* P$_{tetA}$::*pefACDEF* in pWSK29 | 26 |
| p4397 | *tetR* P$_{tetA}$::*bcfABCDEFG* in pWSK29 | 26 |
| p4399 | *tetR* P$_{tetA}$::*stcABC* in pWSK29 | 26 |
| p4400 | *tetR* P$_{tetA}$::*sthABCDE* in pWSK29 | 26 |
| p4401 | *tetR* P$_{tetA}$::*pagN* in pWSK29 | 26 |
| p4402 | *tetR* P$_{tetA}$::*rck* in pWSK29 | 26 |
| p4403 | *tetR* P$_{tetA}$::*misL* in pWSK29 | 26 |
| p4519 | *tetR* P$_{tetA}$::*lpfABCDE* in pWSK29 | 26 |
| p4520 | *tetR* P$_{tetA}$::*shdA* in pWSK29 | 26 |
| p4904 | *tetR* P$_{tetA}$::*hilD* in pWSK29 | 31 |
| p5035 | *tetR* P$_{tetA}$::*sadBA* in pWSK29 | 31 |
| p4318 | p4253 *bapA* ΔBIg1 | 31 |
| p4321 | p4253 *bapA* ΔBIg28 | 31 |
| p4331 | p4253 *bapA* ΔBIg1–ΔBIg28 | 31 |
| p5287 | p4403 *misL* Δaa29–227 | This study |
| p5288 | p4403 *misL* Δaa29–479 | This study |
| p5474 | p4403 *misL* Δaa29–78 | This study |
| p5475 | p4403 *misL* Δaa128–171 | This study |
| p5476 | p4403 *misL* Δaa172–214 | This study |
| p5479 | p4403 *misL* Δaa282–476 | This study |
| p5480 | p4403 *misL* Δaa477–671 | This study |
| p5493 | p4251 *bapA* ΔVCBS1-3 *BCD* | This study |

50 µL of the diluted bacterial solution was performed for 1 h at RT under static conditions (referred to as "static") or for 55 min at RT after centrifugation of 5 min at 500 × *g* (referred to as "forced contact"). After inoculation, leaf discs were washed once with PBS to remove nonbound bacteria. Three leaf discs were transferred to 1.5-mL reaction tubes and washed two further times with PBS by short mixing on a Vortex mixer. Plant tissues were homogenized with a pellet pestle motor in 600 µL of 1% sodium deoxycholate in PBS, and lysates were plated on Mueller-Hinton (MH) agar plates and incubated overnight at 37°C. A noninoculated sample was used in every assay to ensure sterility of *L. sativa* or *V. locusta* plants, as well as a sample inoculated with SR11 Δ12 for comparison to all other strains investigated in this study. Adhesion was determined by the ratio of CFU (in inoculum and homogenate). To adjust variations between biological replicates, levels of adhesion were normalized to adhesion of background strain SR11 Δ12 set to 100%, if indicated.

**Flow cytometry.** For analysis of surface expression of MisL and BapA by flow cytometry, approximately 6 × 10$^8$ bacteria of 3.5-h bacterial subcultures as used for adhesion assays were washed in PBS and then fixed with 3% paraformaldehyde in PBS for 20 min at RT. Bacteria were blocked with 2% goat serum in PBS for 30 min and, afterward, stained with the specific primary antiserum rabbit anti-MisL (1:1,000) or rabbit anti-BapA (1:1,000) overnight at 4°C. Staining with secondary antibody goat anti-rabbit-Alexa488 (1:2,000) was performed for 1 h at RT. Bacteria were measured with an Attune NxT flow cytometer (Thermo Fisher) and analyzed using Attune NxT software version 3.11. Noninduced samples or background strain were used as negative control for gating and set to <1% positive cells. Flow cytometry analysis was done only once for MisL samples and thrice for BapA samples.

**Autoaggregation analysis.** For analysis of autoaggregation of strains expressing MisL and various truncated forms of MisL, subcultures used for the inoculation of *L. sativa* were diluted to 1× 10$^8$ bacteria/mL in PBS. Seven microliters of bacterial suspension was imaged by Zeiss AxioObserver with brightfield microscopy with a 40× objective. Images were recorded by an AxioCam, and data were analyzed using ZEN 2012.

**Statistical analyses.** Statistical significances were calculated by Student's *t* test (two-sided) and performed with Excel (Microsoft Office 2016). Significances are indicated as follows: ns, not significant; *, $P < 0.05$; **, $P < 0.01$; and ***, $P < 0.001$. For adhesion assays with *V. locusta* or *L. sativa*, at least three biological replicates were included unless otherwise noted. The distributions of biological replicates are represented as box plots with mean values and upper and lower quartiles.

## SUPPLEMENTAL MATERIAL

Supplemental material is available online only.

**SUPPLEMENTAL FILE 1**, PDF file, 1.1 MB.

## ACKNOWLEDGMENTS

This work was supported by the Bundesanstalt für Landwirtschaft und Ernährung (BLE) by projects PlantInfect and PlantInfect2, grant 2813HS027. Further support by the DFG by grant SFB 944, project Z, is kindly acknowledged.

We thank the members of the PlantInfect consortium for the fruitful discussion and exchange of reagents. The systematic analyses of GEO data for adhesin expression and critical comments on the manuscript by Janina Noster are gratefully acknowledged. We like to thank Andreas J. Bäumler (UC Davis) and Inigo Lasa for sharing antiserum against MisL and BapA, respectively.

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
