## [Reviewer comments · Microbiology Spectrum]

Microbiology Spectrum

Factors required for adhesion of *Salmonella enterica* serovar Typhimurium to *Lactuca sativa* (lettuce)

Laura Elpers, Lena Lüken, Fabio Lange, and Michael Hensel

Corresponding Author(s): Michael Hensel, Universität Osnabrück

Review Timeline:

Submission Date:	September 2, 2022
Editorial Decision:	October 31, 2022
Revision Received:	November 15, 2022
Editorial Decision:	November 19, 2022
Revision Received:	November 25, 2022
Accepted:	November 29, 2022

Editor: Sandeep Tamber

Reviewer(s): Disclosure of reviewer identity is with reference to reviewer comments included in decision letter(s). The following individuals involved in review of your submission have agreed to reveal their identity: Michelle Danyluk (Reviewer #2)

Transaction Report:

DOI: <https://doi.org/10.1128/spectrum.03436-22>

October 31, 2022

Prof. Michael Hensel
Universität Osnabrück
Abt. Mikrobiologie
Barbarastr. 11
Osnabrück 49076
Germany

Re: Spectrum03436-22 (Factors required for adhesion of *Salmonella enterica* serovar Typhimurium to lettuce (*Lactuca sativa*))

Dear Prof. Michael Hensel:

I apologize again for the delay in the review of your manuscript. The reviewers offer several comments that will improve the readability and applicability of the work.

Link Not Available

Sincerely,

Sandeep Tamber

Journals Department
Editor comments:

- cornsalad has different names depending on world region, please at first mention, include the names mache and lamb's lettuce in parenthesis so readers will know what is being referred to
- The first results sub-section is missing a sub-title
- Line 142 consider replacing the citation with the three references describing the expression conditions for Saf, Sii, PagN

Reviewer comments:

Reviewer #1 (Comments for the Author):

The authors present a very long and detailed analysis of the effect of various Salmonella surface structures on attachment to leafy greens.

My biggest concerns with the manuscript are that (1) the authors only performed their experiments in triplicate and (2) that they present this data on the percent scale.

In my experience three replicates is not sufficient and since the data are not normally distributed on the percent scale, the data should be log transformed and presented on the log percent scale. Since only three replicates were collected this is not enough to observe that the data are highly skewed. I think the authors would find if they repeated an experiment using 20 or 30 replicates, they would find the data would not be normally distributed on the percent adherence scale.

A secondary concern is whether the conditions used to collect the data or actually relevant to real world leafy green contamination or not.

Detailed comments are below.

Page 1:

Content: "lettuce (*Lactuca sativa*)"

Comment: According to the abstract the authors also study corn salad.

Page 2:

Content: "corn salad, rocket salad, or lettuce."

Comment: US readers call rocket arugula. What is corn salad?

Page 2:

Content: "we focused on butterhead lettuce (*Lactuca sativa*) which is contaminated by STM at higher rates compared to corn salad, resulting in prolonged persistence."

Comment: Unclear what point the authors are making. Are they indicating that this is an observation from this current study or is this a past observation they are investigating further?

Page 4:

Content: "Contamination can occur during pre-harvest by the seeds themselves, by irrigation water, or by fertilizers (often based on animal source)."

Comment: Citation needed. I don't know of any evidence that seeds contaminated preharvest leading to outbreaks except in the case of sprouts. Other possible sources would be flooding with contaminated water, wild animal intrusion leading to fecal contamination, or windblown contamination from nearby animal operations.

Page 4:

Content: "compared to corn salad leaves,"

Comment: Corn salad is not known to this reviewer. It might be best the first time a plant food is introduced for the authors to indicate the genus and species name.

Page 7:

Content: "interaction with corn salat"

Comment: Typo

Page 8:

Content: "(Fig. S 1A)."

Comment: Salat typo in figure axis

Page 12:

Content: "To bypass reduced interaction with leaf surfaces possibly resulting from loss of motility, contact was forced by centrifugation(5 min at 500 x g). "

Comment: Is this ever explained in the methods? Are these conditions at all relevant to the real world?

Page 13:

Content: " showed a slight but non-significantly decreased"

Comment: If it's not significantly different it's not "slight" it's not different.

What you can say is that the mean is less than 100%, but the difference is not significant.

Page 16:

Content: "A. J. Bäuml er et al. (45) showed the involvement of Lpf"

Comment: Is it typical to include author initials in these citations? This appears to occur repeatedly throughout the section of the

manuscript. The author should check to make sure this is consistent with the journal style.

Page 16:

Content: "As for Lpf fimbriae, little is known about Sti fimbriae."

Comment: The portion of the sentence before the comma can be deleted and the sentence still communicate to the same idea.

Page 21:

Content: "Lettuce seeds were sterilized with 3% NaClO for 4 min"

Comment: Why? How do the authors know that this treatment does not affect the characteristics of the seeds?

Page 21:

Content: "Adhesion to lettuce and corn salad"

Comment: Figure S 1 mentions static and forced contact. What are the conditions for static and forced contact? They do not appear in the methods as far as I can tell.

Page 21:

Content: "For each condition and strain three leaf discs"

Comment: How big are the disks? Why use discs instead of whole leaves? Can the authors be sure that the inoculum does not contact the cut edge of the disc?

Page 22:

Content: "A non- inoculated sample was used in every assay to ensure sterility of the lettuce or corn salad plants"

Comment: Why is sterility important? In the real world lettuce leaves are not sterile. How can the authors be sure that the procedures for generating sterility did not adversely affect the lettuce leaves so they no longer represent the behavior of real leaves?

Reviewer #2 (Comments for the Author):

This manuscript describes a reductionist, synthetic approach to evaluate different surface structures that contribute to Salmonella Typhimurium's attachment to lettuce. The manuscript is generally well written with sound conclusions. Stylistically, it is very heavy with acronym use; the authors have added unnecessary acronyms (example STM instead of just Salmonella Typhimurium) that make its reading challenging; I would encourage them to eliminate all unnecessary acronyms to increase the readability. For an international audience, I also caution use of the term "salad" as synonymous with a leafy green salad. As defined by the oxford dictionary, a salad is any mixture of raw vegetables; until I googled what a corn salad was by the scientific name of the plant, I believed they were talking about a corn kernel salad, and was highly confused as to why this comparison was being made. As someone who's research focuses on produce safety microbiology, I will not be the only one to make this error prior to seeing the scientific name of the plant. It was complicated by use of the term "salad" throughout the abstract and introduction to mean a leafy green salad, as opposed to a salad mixture of raw vegetables. There also seem to be a number of methods and discussion present in the results section that may better be served moving to the methods or discussion. The introduction is overly long, spending too much time and detail describing all the different surface attachment structures. The results as written do not allow for replication of the work. The importance section lacks an specific discussion of the actual importance of the work performed here.

Specific recommendations to authors are included below.

L23 - STM is an unnecessary acronym in this paper and complicates its readability. While defined here in the abstract, it is not defined in the body of the text.

L26-27 - examples of leafy salads are not necessary.

L28-31 - a description of previous findings in the abstract is not necessary, this section can be deleted.

L34 - suggest adding another sentence on methods, to describe that all results and analysis are carried out through a normalization and comparison process to strain SR11 delta12.

L40 - this first sentence of the importance is not necessary, transmission of Salmonella through non animal sources is a well established fact.

L44 - what has to be part of further research does not describe what the importance of this work is, delete" and have to be part of further research"

L44-46 - this section should describe why it is important to identify the adhesive structures involved in binding to lettuce leaves.

This sentence should be added to the abstract as it forms a more concise summary of the work than is described there.

L54 - reference 2 appears to be an EFSA opinion paper, are there other references to describe the increasing number of Salmonella associated with Fresh produce?

L55 - i.e. examples of animal products and fresh produce are not necessary.

L57 - Is there a reference for seeds contamination leading to any outbreak from fresh produce other than sprouts? Pre-harvest factors for contamination expand well beyond just irrigation, seeds and fertilizers to all agricultural water use, intrusion by domestic and wild animals, employees, equipment and tools used. Post harvest contamination includes factors beyond improper hygiene and washing/cutting. Suggest expanding and adding a reference.

L60 - please cite directly an example where Salmonella was found to replicate on the surface of a fruit or vegetable under non-laboratory conditions. Is replication necessary for further spread?

L62-69 - see above comment re. salad. I believe as written in this section that the term salad is specifically referring to a leafy green salad. I suggest being specific here that you are referencing leaf species, not all salad types, as salad's general meaning is a mixture of raw vegetables. This comment holds true throughout the paper.

L70 - please add the references to these studies investigating other adhesion mechanisms.

L73 - please specify that in this case internalization is through stomata, not through the general leaf surface as is implied.

L78 - 117 - this section describing adhesion mechanisms related to different structures should be significantly shortened to give just important highlights necessary to frame the work being done here; a further description of critical factors found to be important here can be included in the discussion.

L118- this section describing the Tet-On system is not necessary in the introduction. It could move to the methods if you wish to include.

L159 - unclear from my reading of the methods why more than 11 samples could not be performed at one time

L166 - cognate ligands maybe absent on lettuce leaves - this was not mentioned/expanded upon in the discussion. Are there references to further this finding?

L205-207 - this should move to the discussion

L221-231 - this should move to methods.

L245-248 - this should move to discussion and methods.

L287/fig 7 - it is unclear what the purpose of Figure 7 is. This is its only reference. Please expand on why it is here, or reference it more in the text.

L301-304 - I can understand why you are hypothesizing this direct motility, but I think it should be framed more as a hypothesis, as you did not do direct microscopy to determine if in fact this is what was happening on your samples.

L309 - how long does it take to do this swimming and searching? How far is it possible for Salmonella to swim? How would the density of preferred sites on the leaves impact the attachment? Are there articles that reference the density of preferred sites on different types of leafy greens?

L314-315 - suggest deleting this reference to virulence in mammalian cells. Salmonella is not virulent to plant cells, so this adds nothing to the findings.

L321-322 - linking studies on Arabidopsis and Romaine attachment to its important on "all plants", which implies all plant tissues, is a bit of a stretch. Consider deleting or softening the language here.

L381 - is the *S. enterica* lifestyle being spoken of here in mammalian cells? Does *S. enterica* have a lifestyle on plants?

L 415-416 - delete this last sentence.

Discussion in general - I think what might be missing here is something about the bigger picture of why this work is important - you start your introduction speaking about increasing outbreaks from produce, and multiple ways plants can be contaminated. Could you wrap this up by speaking back to these points, and how better understanding the initial attachment could help in preventing contamination?

L429 - do we know what the original sources of the Salmonella Typhimurium strains are? While beyond the scope of this work, I'm curious if the same results would hold true for recent outbreak strains of Salmonella.

L427 - How large are the leaf disks?

L476 - for all replicates you had exactly the same inoculum level? There was no variation in here. if you used a lower inoculation level, does this impact results?

L476-477 - did 50 ul cover the entire surface of the leaf disk. How did you ensure in this experimental design the attachment was to the intact surface of the leaf and not the cut edges of the leaf. If you could not, how might attachment to cut surfaces be different from attachment to intact surfaces on the leaf, and how may this impact the interpretation of all your results.

L477 - why one hour? What are RT conditions in your laboratory in terms of temperature and relative humidity? How would temperature impact or change the results?

L478 - how were leaf disks washed? How much washing matrix was used? About how long was short mixing on the vortex?

L484 - 488 - do you ever report the actual concentrations in your samples? I'm curious to what % actually attach, when not normalized. In general how representative is this inoculation procedure when compared to how Salmonella may be contaminating plants in the field? How could this inoculation style impact your results when compared to natural contamination?

Staff Comments:

Preparing Revision Guidelines

Please return the manuscript within 60 days; if you cannot complete the modification within this time period, please contact me. If you do not wish to modify the manuscript and prefer to submit it to another journal, please notify me of your decision immediately so that the manuscript may be formally withdrawn from consideration by Microbiology Spectrum.

We thank the reviewers for their comments that helped us to improve the presentation of the data in the manuscript. Please find below a point-by-point response to the comments. Our response is indicated by red font.

Please note that we replaced the commercial names of salads by the systematic binominal designations to avoid confusion. We also substantially revised the discussion to improve clarity.

Editor comments:

- cornsalad has different names depending on world region, please at first mention, include the names mache and lamb's lettuce in parenthesis so readers will know what is being referred to

Response: We included the other names for corn salad, and used the binominal name of both salad species throughout.

- The first results sub-section is missing a sub-title

Response: We added a sub-title.

- Line 142 consider replacing the citation with the three references describing the expression conditions for Saf, Sij, PagN

The citation was moved as suggested.

Reviewer comments:

Reviewer #1 (Comments for the Author):

The authors present a very long and detailed analysis of the effect of various Salmonella surface structures on attachment to leafy greens.

My biggest concerns with the manuscript are that (1) the authors only performed their experiments in triplicate and (2) that they present this data on the percent scale.

In my experience three replicates is not sufficient and since the data are not normally distributed on the percent scale, the data should be log transformed and presented on the log percent scale. Since only three replicates were collected this is not enough to observe that the data are highly skewed. I think the authors would find if they repeated an experiment using 20 or 30 replicates, they would find the data would not be normally distributed on the percent adherence scale.

Response: An important aspect of the work is the direct comparison between adhesion to corn salad and to lettuce. The experimental conditions used in the present manuscript are identical to the conditions applied on the previous study by Elpers et al., 2020. The only difference is the host for adhesion, here lettuce, and the subsequent analyses of the factors required for adhesion to lettuce. Changing the presentation of the data would severely complicate the comparison between the salad species. Please appreciate that in each experiment, 3 leaf disks were used for infection per condition (technical replicates). We also have established a synthetic system to analyze adhesion under

defined experimental conditions, and distinct from field conditions. This allows us to draw conclusions from three biological replicates.

A secondary concern is whether the conditions used to collect the data are actually relevant to real world leafy green contamination or not.

Response: As stated in the description of our approach, we use a reductionist, synthetic approach to identify factors that could mediate adhesion of *Salmonella* to salad leaves. We do not claim to transfer the data generated here directly to the given cultivation and environmental conditions, as well as possible contaminations. In this work, we show basic investigations using the reductionist synthetic approach to identify possible factors of *Salmonella enterica* that enable initial adhesion and subsequent colonization of leafy green salads. With the knowledge gained from this study, further investigations can now be carried out into the importance of individual adhesive structures in the adhesion and colonization of lettuce leaves under native cultivation conditions. In any case, the environmental factors (temperature, UV light, type of irrigation) and the microbiome of the plant should be taken into account.

Detailed comments are below.

Page 1:

Content: "lettuce (*Lactuca sativa*)"

Comment: According to the abstract the authors also study corn salad.

Response: Yes, in this study we also worked with corn salad (*Valerianella locusta*). But since we only performed one experiment with corn salad (only shown in the supplement) and focused on lettuce (*Lactuca sativa*), we did not include corn salad in the title.

Page 2:

Content: "corn salad, rocket salad, or lettuce."

Comment: US readers call rocket arugula. What is corn salad?

Response: The list of salads deleted in response to reviewer 2. Alternative names of *Valerianella locusta* defined in line 64.

Page 2:

Content: "we focused on butterhead lettuce (*Lactuca sativa*) which is contaminated by STM at higher rates compared to corn salad, resulting in prolonged persistence."

Comment: Unclear what point the authors are making. Are they indicating that this is an observation from this current study or is this a past observation they are investigating further?

Response: We rearranged the section to point out that we previously investigated the impact of adhesive structures of *Salmonella enterica* to corn salad leaves. A study of a collaboration partner, showed longer persistence of *Salmonella* on lettuce compared to corn salad (Jechalke S, Schierstaedt J, Becker M, Flemer B, Grosch R, Smalla K, Schikora A. 2019. *Salmonella* establishment in agricultural soil and colonization of crop plants depend on soil type and plant species. *Front Microbiol* 10:967.) which is mentioned in the introduction section. In the following we observed higher adhesion rates of

Salmonella to lettuce leaves in comparison to corn salad leaves (shown in the supplement) thus we decided to also investigate the impact of adhesive structures of *Salmonella enterica* to lettuce leaves.

Page 4:

Content: "Contamination can occur during pre-harvest by the seeds themselves, by irrigation water, or by fertilizers (often based on animal source)."

Comment: Citation needed. I don't know of any evidence that seeds contaminated preharvest leading to outbreaks except in the case of sprouts. Other possible sources would be flooding with contaminated water, wild animal intrusion leading to fecal contamination, or windblown contamination from nearby animal operations.

Response: We did not state that outbreaks were associated with contaminated seeds, but several studies showed that contamination of seeds can result in plants with detectable numbers of the respective bacteria (citation added). We added the suggested additional possible sources of contaminations.

Page 4:

Content: "compared to corn salad leaves,"

Comment: Corn salad is not known to this reviewer. It might be best the first time a plant food is introduced for the authors to indicate the genus and species name.

Response: We added the genus and species name.

Page 7:

Content: "interaction with corn salat"

Comment: Typo

Response: Changed accordingly.

Page 8:

Content: "(Fig. S 1A)."

Comment: Salat typo in figure axis

Response: Changed accordingly.

Page 12:

Content: "To bypass reduced interaction with leaf surfaces possibly resulting from loss of motility, contact was forced by centrifugation (5 min at 500 x g). "

Comment: Is this ever explained in the methods? Are these conditions at all relevant to the real world?

Response: The centrifugation step during the adhesion assay to leafy green salad leaves to ensure forced contact is explained in the Material and Methods section "Adhesion to lettuce and corn salad". Of course, in the "real world" no centrifugation of bacteria to the leave surface will occur. However we centrifugated to circumvent the fact, that some *Salmonella* mutants are non-motile but still harbors the flagella on the bacterial surface being unable to get in contact with the leave surface by themselves. With this experimental setting we were able to discriminate between adhesion to lettuce leaves by motility and adhesion to lettuce leaves by the flagella as an adhesive structure.

Page 13:

Content: " showed a slight but non-significantly decreased"

Comment: If it's not significantly different it's not "slight" it's not different.

What you can say is that the mean is less than 100%, but the difference is not significant.

Response: Changed accordingly.

Page 16:

Content: "A. J. Bäumlér et al. (45) showed the involvement of Lpf"

Comment: Is it typical to include author initials in these citations? This appears to occur repeatedly throughout the section of the manuscript. The author should check to make sure this is consistent with the journal style.

Response: In the revised version we changed citations to ASM style. The initial submission was format neutral and submitted via a repository.

Page 16:

Content: "As for Lpf fimbriae, little is known about Sti fimbriae."

Comment: The portion of the sentence before the comma can be deleted and the sentence still communicate to the same idea.

Response: Changed accordingly.

Page 21:

Content: "Lettuce seeds were sterilized with 3% NaClO for 4 min"

Comment: Why? How do the authors know that this treatment does not affect the characteristics of the seeds?

Response: Surface sterilization of seeds with NaClO is a standardized method in the field to prevent any microbial contaminations during plant growth under sterile conditions.

Page 21:

Content: "Adhesion to lettuce and corn salad"

Comment: Figure S 1 mentions static and forced contact. What are the conditions for static and forced contact? They do not appear in the methods as far as I can tell.

Response: We added a distinct explanation for the designations static and forced contact in the Material and Methods section "Adhesion to lettuce and corn salad".

Page 21:

Content: "For each condition and strain three leaf discs"

Comment: How big are the disks? Why use discs instead of whole leaves? Can the authors be sure that the inoculum does not contact the cut edge of the disc?

Response: The leaf discs had a diameter of 8 mm. We ensured that the leaf discs were not floating and prevented contact of bacteria with the cut edges by using inlays of stainless steel with an inner

diameter of 6 mm placed on the leaf discs. Leaf discs with the distinct inlays ensured a repeatable area of leaf being colonized by a specific amount of bacteria. We added a further reference describing the used method step by step with detailed illustration.

Page 22:

Content: "A non- inoculated sample was used in every assay to ensure sterility of the lettuce or corn salad plants"

Comment: Why is sterility important? In the real world lettuce leaves are not sterile. How can the authors be sure that the procedures for generating sterility did not adversely affect the lettuce leaves so they no longer represent the behavior of real leaves?

Response: Sterility of the plants is part of our synthetic reductionist approach to eliminate any unpredictable variations, such as diverse and changing microbiomes. Since the microbiome is not static and is individual for every plant we could not ensure that changes in the adhesion rates only depends on the respective adhesive structure we are investigating. We are aware of the fact that adhesion and following colonization of lettuce leaves by Salmonella will be affected by "real world" conditions like the microbiome, changing temperatures or UV light. With the results obtained in this study we are looking for following up studies to reveal the impact of the adhesive structures under "real world" conditions.

Reviewer #2 (Comments for the Author):

This manuscript describes a reductionist, synthetic approach to evaluate different surface structures that contribute to Salmonella Typhimurium's attachment to lettuce. The manuscript is generally well written with sound conclusions. Stylistically, it is very heavy with acronym use; the authors have added unnecessary acronyms (example STM instead of just Salmonella Typhimurium) that make its reading challenging; I would encourage them to eliminate all unnecessary acronyms to increase the readability. For an international audience, I also caution use of the term "salad" as synonymous with a leafy green salad. As defined by the oxford dictionary, a salad is any mixture of raw vegetables; until I googled what a corn salad was by the scientific name of the plant, I believed they were talking about a corn kernel salad, and was highly confused as to why this comparison was being made. As someone who's research focuses on produce safety microbiology, I will not be the only one to make this error prior to seeing the scientific name of the plant. It was complicated by use of the term "salad" throughout the abstract and introduction to mean a leafy green salad, as opposed to a salad mixture of raw vegetables. There also seem to be a number of methods and discussion present in the results section that may better be served moving to the methods or discussion. The introduction is overly long, spending too much time and detail describing all the different surface attachment structures. The results as written do not allow for replication of the work. The importance section lacks an specific discussion of the actual importance of the work performed here. Specific recommendations to authors are included below.

L23 - STM is an unnecessary acronym in this paper and complicates its readability. While defined here in the abstract, it is not defined in the body of the text.

Response: The acronym STM is defined in the section "Introduction" in line 68/69. ASM style would demand *S. enterica* serovar Typhimurium, not *Salmonella* Typhimurium or *S. Typhimurium*, which

would severely inflate the text compared to use of STM. We find it very helpful and space saving to use STM instead of repeating *S. enterica* serovar Typhimurium.

L26-27 - examples of leafy salads are not necessary.

Response: deleted

L28-31 - a description of previous findings in the abstract is not necessary, this section can be deleted.

Response: Changed accordingly.

L34 - suggest adding another sentence on methods, to describe that all results and analysis are carried out through a normalization and comparison process to strain SR11 delta12.

Response: In the results section the normalization of all experiments analyzing STM adhesion to leafy green salad leaves to strain SR11 delta12 is well described. We do not see the need for highlighting this aspect in the abstract section.

L40 - this first sentence of the importance is not necessary, transmission of Salmonella through non animal sources is a well established fact.

Response: Sentence deleted.

L44 - what has to be part of further research does not describe what the importance of this work is, delete" and have to be part of further research"

Response: Changed accordingly.

L44-46 - this section should describe why it is important to identify the adhesive structures involved in binding to lettuce leaves. This sentence should be added to the abstract as it forms a more concise summary of the work than is described there.

Response: We revised Abstract and Importance sections accordingly

L54 - reference 2 appears to be an EFSA opinion paper, are there other references to describe the increasing number of Salmonella associated with Fresh produce?

Response: We added another reference.

L55 - i.e. examples of animal products and fresh produce are not necessary.

Response: Changed accordingly.

L57 - Is there a reference for seeds contamination leading to any outbreak from fresh produce other than sprouts? Pre-harvest factors for contamination expand well beyond just irrigation, seeds and fertilizers to all agricultural water use, intrusion by domestic and wild animals, employees,

equipment and tools used. Post harvest contamination includes factors beyond improper hygiene and washing/cutting. Suggest expanding and adding a reference.

Response: We added further references.

L60 - please cite directly an example where Salmonella was found to replicate on the surface of a fruit or vegetable under non-laboratory conditions. Is replication necessary for further spread?

Response: We cited a further study and review. We did not find studies on replication on fruits or vegetables under open field conditions but at least under laboratory conditions trying to mimic normal agricultural practices.

L62-69 - see above comment re. salad. I believe as written in this section that the term salad is specifically refereeing to a leafy green salad. I suggest being specific here that you are referencing leaf species, not all salad types, as salad's general meaning is a mixture of raw vegetables. This comment holds true throughout the paper.

Response: We changed the designation "salad" to "leafy green salad" throughout the paper.

L70 - please add the references to these studies investigating other adhesion mechanisms.

Response: We rewrote the sentence.

L73 - please specify that in this case internalization is through stomata, not through the general leaf surface as is implied.

Response: Changed accordingly.

L78 - 117 - this section describing adhesion mechanisms related to different structures should be significantly shortened to give just important highlights necessary to frame the work being done here; a further description of critical factors found to be important here can be included in the discussion.

Response: We have shortened the section.

L118- this section describing the Tet-On system is not necessary in the introduction. It could move to the methods if you wish to include.

Response: We shortened the paragraph.

L159 - unclear from my reading of the methods why more than 11 samples could not be performed at one time

Response: Preparation and handling of the lettuce leaf discs takes a certain amount of time. If more than 11 samples are handled at the same time, the lettuce leaf discs will be affected (e.g. degeneration will start since leaves are separated from the plant itself).

L166 - cognate lignands maybe absent on lettuce leaves - this was not mentioned/expanded upon in

the discussion. Are there references to further this finding?

Response: We added a section to the discussion.

L205-207 - this should move to the discussion

Response: We found this information better placed in the introduction.

L221-231 - this should move to methods.

Response: The information given here does not correspond to a methodological description, but explains the structure of the respective adhesin, which is important for the understanding of the following results concerning truncated forms of the adhesin.

L245-248 - this should move to discussion and methods.

Response: Without the explanations given here, we believe it is unlikely that the reader will understand the results that follow.

L287/fig 7 - it is unclear what the purpose of Figure 7 is. This is its only reference. Please expand on why it is here, or reference it more in the text.

Response: Figure 7 only serves as a visual summary of the results obtained in this work thus it is only mentioned one time. The summary serves to facilitate the comparison of the collected results regarding lettuce leaves in this study, but also the comparison to the previous study on the adherence factors of STM on lamb's lettuce leaves (Elpers et al 2020).

L301-304 - I can understand why you are hypothesing this direct motility, but I think it should be framed more as a hypothesis, as you did not do direct microscopy to determine if infact this is what was happening on your samples.

Response: Changed accordingly.

L309 - how long does it take to do this swimming and searching? How far is it possible for Salmonella to swim? How would the density of preferred sites on the leaves impact the attachment? Are there articles that reference the density of preferred sites on different types of leafy greens?

Response: Citations added for studies that support such role in interaction with mammalian cell.

L314-315 - suggest deleting this reference to virulence in mammalian cells. Salmonella is not virulent to plant cells, so this adds nothing to the findings.

Response: We have rewritten the section.

L321-322 - linking studies on Arabidopsis and Romaine attachment to its important on "all plants", which implies all plant tissues, is a bit of a stretch. Consider deleting or softening the language here.

Response: Changed accordingly.

L381 - is the *S. enterica* lifestyle being spoken of here in mammalian cells? Does *S. enterica* have a lifestyle on plants?

Response: We rewrote the sentence. Studies showed that BapA can be involved in the lifestyle during infection of mammalian cells, as well as outside the host, e.g. during biofilm formation on abiotic surfaces.

L 415-416 - delete this last sentence.

Response: Changed accordingly.

Discussion in general - I think what might be missing here is something about the bigger picture of why this work is important - you start your introduction speaking about increasing outbreaks from produce, and multiple ways plants can be contaminated. Could you wrap this up by speaking back to these points, and how better understanding the initial attachment could help in preventing contamination?

Response: We included a further section in the discussion about possible anti-adhesive structures during agricultural growth.

L429 - do we know what the original sources of the *Salmonella* Typhimurium strains are? While beyond the scope of this work, I'm curious if the same results would hold true for recent outbreak strains of *Salmonella*.

Response: The here used *Salmonella enterica* serovar Typhimurium strain NCTC 12023 (also called ATCC 14028) was originally isolated from pooled heart and liver samples from 4-week old chickens in 1960 (<https://doi.org/10.1128/JB.01233-09>). We are aware that STM may have evolved. However, yet we do not know any further adhesive structures encoded by STM which might impact adhesion to leafy green salad.

L427 - How large are the leaf disks?

Response: The leaf discs had a diameter of 8 mm. We ensured that the leaf discs were not floating and prevented contact of bacteria with the cut edges by using inlays of stainless steel with an inner diameter of 6 mm placed on the leaf discs. Leaf discs with the distinct inlays ensured a repeatable area of leaf being colonized by specific amounts of bacteria. We added a further reference that describes the applied methods step by step with detailed illustration.

L476 - for all replicates you had exactly the same inoculum level? There was no variation in here. if you used a lower inoculation level, does this impact results?

Response: There was not variation here regarding the inoculum. We did not check for the impact of lower inoculation levels of STM on the adhesion to lettuce leaves.

L476-477 - did 50 ul cover the entire surface of the leaf disk. How did you ensure in this experimental design the attachment was to the intact surface of the leaf and not the cut edges of the leaf. If you

could not, how might attachment to cut surfaces be different from attachment to intact surfaces on the leaf, and how may this impact the interpretation of all your results.

Response: The leaf discs had a diameter of 8 mm. We ensured that the leaf discs were not floating and prevented contact of bacteria with the cut edges by using inlays of stainless steel with an inner diameter of 6 mm placed on the leaf discs. Leaf discs with the distinct inlays ensured a repeatable area of leaf being colonized by a specific amount of bacteria. Using 50 μl of the inoculum covered the area of a leaf disc with 6 mm diameter. We added a further reference describing the applied methods step by step with detailed illustration.

L477 - why one hour? What are RT conditions in your laboratory in terms of temperature and relative humidity? How would temperature impact or change the results?

Response: 1 hour incubation allows the determination of adherent bacteria without affecting the plant material severely. Room temperature means 21 °C. Since inoculation occurred by bacteria in buffer, the relative humidity appears of limited relevance of this assay. We tried once to affect STM adhesion to leaf green leaves by changing the temperature. This resulted in decreased adhesion at lower temperatures.

L478 - how were leaf disks washed? How much washing matrix was used? About how long was short mixing on the vortex?

Response: Leaf discs were washed once with 200 μl PBS when inlays were still placed on the leaf discs. Then after collection of leaf discs in tubes, leaf discs were washed twice with 1 ml PBS. Short mixing on the vortex equals 2 seconds. We added a further reference describing the used method step by step with detailed illustration.

L484 - 488 - do you ever report the actual concentrations in your samples? I'm curious to what % actually attach, when not normalized. In general how representative is this inoculation procedure when compared to how Salmonella may be contaminating plants in the field? How could this inoculation style impact your results when compared to natural contamination?

Response: In the Materials and Method section we report using 50 μl of an inoculum of 5.6×10^7 bacteria/ml. The supplementary figure 1A shows adhesions rates in % of the inoculum for STM to corn salad leaves and to lettuce leaves. Therefore, the actual adhered number of bacteria to the leaf disc can be calculated easily. Here we do not want to investigate the possible routes of contamination, but rather the influence of the various adhesive structures for binding STM to leaves of leafy green salads. It is obvious that the inoculation style has an influence on the further colonisation of STM. Natural contamination conditions include an enormous number of possibilities (irrigation, temperature, UV-light, microbiome, "media" in which the STM are present; ...) which could only be tested to a limited extent in this experimental set-up.

November 19, 2022

Prof. Michael Hensel
Universität Osnabrück
Abt. Mikrobiologie
Barbarastr. 11
Osnabrück 49076
Germany

Re: Spectrum03436-22R1 (Factors required for adhesion of *Salmonella enterica* serovar Typhimurium to *Lactuca sativa* (lettuce))

Dear Prof. Michael Hensel:

Thank you for submitting your manuscript to Microbiology Spectrum. As you will see your paper is very close to acceptance. Please modify the manuscript along the lines I have recommended. As these revisions are quite minor, I expect that you should be able to turn in the revised paper in less than 30 days, if not sooner. If your manuscript was reviewed, you will find the reviewers' comments below.

When submitting the revised version of your paper, please provide (1) point-by-point responses to the issues raised by the reviewers as file type "Response to Reviewers," not in your cover letter, and (2) a PDF file that indicates the changes from the original submission (by highlighting or underlining the changes) as file type "Marked Up Manuscript - For Review Only". Please use this link to submit your revised manuscript. Detailed instructions on submitting your revised paper are below.

Link Not Available

Sincerely,

Sandeep Tamber

Editor comments:

A few minor comments to improve clarity:

L29 -30: As written it is unclear. Please change - suggestion follows: We focused on butterhead lettuce (*Lactuca sativa*) to which *S. enterica* serovar Typhimurium adheres to at higher rates compared to *Valerianella locusta*,

L56: mention of contaminated seeds, and L60: mention of replication in plants. I agree with the reviewers, reading these statements in the context of the introduction implies that these events can lead to illnesses. However, this is not backed by the science. The citations refer to lab studies. My suggestion would be to remove the mention of the seeds and in plant replication or state very clearly that the evidence is based on lab studies and not from in field or outbreak investigations.

Once these changes are made, I will be very happy to accept this paper for publication.

Preparing Revision Guidelines

To submit your modified manuscript, log onto the eJP submission site at <https://spectrum.msubmit.net/cgi-bin/main.plex>. Go to Author Tasks and click the appropriate manuscript title to begin the revision process. The information that you entered when you first submitted the paper will be displayed. Please update the information as necessary. Here are a few examples of required

updates that authors must address:

Please return the manuscript within 60 days; if you cannot complete the modification within this time period, please contact me. If you do not wish to modify the manuscript and prefer to submit it to another journal, please notify me of your decision immediately so that the manuscript may be formally withdrawn from consideration by Microbiology Spectrum.

We thank the reviewers for their comments that helped us to improve the presentation of the data in the manuscript. Please find below a point-by-point response to the comments. Our response is indicated by red font.

Please note that we replaced the commercial names of salads by the systematic binomial designations to avoid confusion. We also substantially revised the discussion to improve clarity.

Editor comments:

- cornsalad has different names depending on world region, please at first mention, include the names mache and lamb's lettuce in parenthesis so readers will know what is being referred to

Response: We included the other names for corn salad, and used the binomial name of both salad species throughout.

- The first results sub-section is missing a sub-title

Response: We added a sub-title.

- Line 142 consider replacing the citation with the three references describing the expression conditions for Saf, Sij, PagN

The citation was moved as suggested.

Reviewer comments:

Reviewer #1 (Comments for the Author):

The authors present a very long and detailed analysis of the effect of various Salmonella surface structures on attachment to leafy greens.

My biggest concerns with the manuscript are that (1) the authors only performed their experiments in triplicate and (2) that they present this data on the percent scale.

In my experience three replicates is not sufficient and since the data are not normally distributed on the percent scale, the data should be log transformed and presented on the log percent scale. Since only three replicates were collected this is not enough to observe that the data are highly skewed. I think the authors would find if they repeated an experiment using 20 or 30 replicates, they would find the data would not be normally distributed on the percent adherence scale.

Response: An important aspect of the work is the direct comparison between adhesion to corn salad and to lettuce. The experimental conditions used in the present manuscript are identical to the conditions applied on the previous study by Elpers et al., 2020. The only difference is the host for adhesion, here lettuce, and the subsequent analyses of the factors required for adhesion to lettuce. Changing the presentation of the data would severely complicate the comparison between the salad species. Please appreciate that in each experiment, 3 leaf disks were used for infection per condition (technical replicates). We also have established a synthetic system to analyze adhesion under

defined experimental conditions, and distinct from field conditions. This allows us to draw conclusions from three biological replicates.

A secondary concern is whether the conditions used to collect the data are actually relevant to real world leafy green contamination or not.

Response: As stated in the description of our approach, we use a reductionist, synthetic approach to identify factors that could mediate adhesion of *Salmonella* to salad leaves. We do not claim to transfer the data generated here directly to the given cultivation and environmental conditions, as well as possible contaminations. In this work, we show basic investigations using the reductionist synthetic approach to identify possible factors of *Salmonella enterica* that enable initial adhesion and subsequent colonization of leafy green salads. With the knowledge gained from this study, further investigations can now be carried out into the importance of individual adhesive structures in the adhesion and colonization of lettuce leaves under native cultivation conditions. In any case, the environmental factors (temperature, UV light, type of irrigation) and the microbiome of the plant should be taken into account.

Detailed comments are below.

Page 1:

Content: "lettuce (*Lactuca sativa*)"

Comment: According to the abstract the authors also study corn salad.

Response: Yes, in this study we also worked with corn salad (*Valerianella locusta*). But since we only performed one experiment with corn salad (only shown in the supplement) and focused on lettuce (*Lactuca sativa*), we did not include corn salad in the title.

Page 2:

Content: "corn salad, rocket salad, or lettuce."

Comment: US readers call rocket arugula. What is corn salad?

Response: The list of salads deleted in response to reviewer 2. Alternative names of *Valerianella locusta* defined in line 64.

Page 2:

Content: "we focused on butterhead lettuce (*Lactuca sativa*) which is contaminated by STM at higher rates compared to corn salad, resulting in prolonged persistence."

Comment: Unclear what point the authors are making. Are they indicating that this is an observation from this current study or is this a past observation they are investigating further?

Response: We rearranged the section to point out that we previously investigated the impact of adhesive structures of *Salmonella enterica* to corn salad leaves. A study of a collaboration partner, showed longer persistence of *Salmonella* on lettuce compared to corn salad (Jechalke S, Schierstaedt J, Becker M, Flemer B, Grosch R, Smalla K, Schikora A. 2019. *Salmonella* establishment in agricultural soil and colonization of crop plants depend on soil type and plant species. Front Microbiol 10:967.) which is mentioned in the introduction section. In the following we observed higher adhesion rates of

Salmonella to lettuce leaves in comparison to corn salad leaves (shown in the supplement) thus we decided to also investigate the impact of adhesive structures of *Salmonella enterica* to lettuce leaves.

Page 4:

Content: "Contamination can occur during pre-harvest by the seeds themselves, by irrigation water, or by fertilizers (often based on animal source)."

Comment: Citation needed. I don't know of any evidence that seeds contaminated preharvest leading to outbreaks except in the case of sprouts. Other possible sources would be flooding with contaminated water, wild animal intrusion leading to fecal contamination, or windblown contamination from nearby animal operations.

Response: We did not state that outbreaks were associated with contaminated seeds, but several studies showed that contamination of seeds can result in plants with detectable numbers of the respective bacteria (citation added). We added the suggested additional possible sources of contaminations.

Page 4:

Content: "compared to corn salad leaves,"

Comment: Corn salad is not known to this reviewer. It might be best the first time a plant food is introduced for the authors to indicate the genus and species name.

Response: We added the genus and species name.

Page 7:

Content: "interaction with corn salat"

Comment: Typo

Response: Changed accordingly.

Page 8:

Content: "(Fig. S 1A)."

Comment: Salat typo in figure axis

Response: Changed accordingly.

Page 12:

Content: "To bypass reduced interaction with leaf surfaces possibly resulting from loss of motility, contact was forced by centrifugation (5 min at 500 x g). "

Comment: Is this ever explained in the methods? Are these conditions at all relevant to the real world?

Response: The centrifugation step during the adhesion assay to leafy green salad leaves to ensure forced contact is explained in the Material and Methods section "Adhesion to lettuce and corn salad". Of course, in the "real world" no centrifugation of bacteria to the leave surface will occur. However we centrifugated to circumvent the fact, that some *Salmonella* mutants are non-motile but still harbors the flagella on the bacterial surface being unable to get in contact with the leave surface by themselves. With this experimental setting we were able to discriminate between adhesion to lettuce leaves by motility and adhesion to lettuce leaves by the flagella as an adhesive structure.

Page 13:

Content: " showed a slight but non-significantly decreased"

Comment: If it's not significantly different it's not "slight" it's not different.

What you can say is that the mean is less than 100%, but the difference is not significant.

Response: Changed accordingly.

Page 16:

Content: "A. J. Bäumler et al. (45) showed the involvement of Lpf"

Comment: Is it typical to include author initials in these citations? This appears to occur repeatedly throughout the section of the manuscript. The author should check to make sure this is consistent with the journal style.

Response: In the revised version we changed citations to ASM style. The initial submission was format neutral and submitted via a repository.

Page 16:

Content: "As for Lpf fimbriae, little is known about Sti fimbriae."

Comment: The portion of the sentence before the comma can be deleted and the sentence still communicate to the same idea.

Response: Changed accordingly.

Page 21:

Content: "Lettuce seeds were sterilized with 3% NaClO for 4 min"

Comment: Why? How do the authors know that this treatment does not affect the characteristics of the seeds?

Response: Surface sterilization of seeds with NaClO is a standardized method in the field to prevent any microbial contaminations during plant growth under sterile conditions.

Page 21:

Content: "Adhesion to lettuce and corn salad"

Comment: Figure S 1 mentions static and forced contact. What are the conditions for static and forced contact? They do not appear in the methods as far as I can tell.

Response: We added a distinct explanation for the designations static and forced contact in the Material and Methods section "Adhesion to lettuce and corn salad".

Page 21:

Content: "For each condition and strain three leaf discs"

Comment: How big are the disks? Why use discs instead of whole leaves? Can the authors be sure that the inoculum does not contact the cut edge of the disc?

Response: The leaf discs had a diameter of 8 mm. We ensured that the leaf discs were not floating and prevented contact of bacteria with the cut edges by using inlays of stainless steel with an inner

diameter of 6 mm placed on the leaf discs. Leaf discs with the distinct inlays ensured a repeatable area of leaf being colonized by a specific amount of bacteria. We added a further reference describing the used method step by step with detailed illustration.

Page 22:

Content: "A non- inoculated sample was used in every assay to ensure sterility of the lettuce or corn salad plants"

Comment: Why is sterility important? In the real world lettuce leaves are not sterile. How can the authors be sure that the procedures for generating sterility did not adversely affect the lettuce leaves so they no longer represent the behavior of real leaves?

Response: Sterility of the plants is part of our synthetic reductionist approach to eliminate any unpredictable variations, such as diverse and changing microbiomes. Since the microbiome is not static and is individual for every plant we could not ensure that changes in the adhesion rates only depends on the respective adhesive structure we are investigating. We are aware of the fact that adhesion and following colonization of lettuce leaves by Salmonella will be affected by "real world" conditions like the microbiome, changing temperatures or UV light. With the results obtained in this study we are looking for following up studies to reveal the impact of the adhesive structures under "real world" conditions.

Reviewer #2 (Comments for the Author):

This manuscript describes a reductionist, synthetic approach to evaluate different surface structures that contribute to Salmonella Typhimurium's attachment to lettuce. The manuscript is generally well written with sound conclusions. Stylistically, it is very heavy with acronym use; the authors have added unnecessary acronyms (example STM instead of just Salmonella Typhimurium) that make its reading challenging; I would encourage them to eliminate all unnecessary acronyms to increase the readability. For an international audience, I also caution use of the term "salad" as synonymous with a leafy green salad. As defined by the oxford dictionary, a salad is any mixture of raw vegetables; until I googled what a corn salad was by the scientific name of the plant, I believed they were talking about a corn kernel salad, and was highly confused as to why this comparison was being made. As someone who's research focuses on produce safety microbiology, I will not be the only one to make this error prior to seeing the scientific name of the plant. It was complicated by use of the term "salad" throughout the abstract and introduction to mean a leafy green salad, as opposed to a salad mixture of raw vegetables. There also seem to be a number of methods and discussion present in the results section that may better be served moving to the methods or discussion. The introduction is overly long, spending too much time and detail describing all the different surface attachment structures. The results as written do not allow for replication of the work. The importance section lacks an specific discussion of the actual importance of the work performed here. Specific recommendations to authors are included below.

L23 - STM is an unnecessary acronym in this paper and complicates its readability. While defined here in the abstract, it is not defined in the body of the text.

Response: The acronym STM is defined in the section "Introduction" in line 68/69. ASM style would demand *S. enterica* serovar Typhimurium, not *Salmonella* Typhimurium or *S. Typhimurium*, which

would severely inflate the text compared to use of STM. We find it very helpful and space saving to use STM instead of repeating *S. enterica* serovar Typhimurium.

L26-27 - examples of leafy salads are not necessary.

Response: deleted

L28-31 - a description of previous findings in the abstract is not necessary, this section can be deleted.

Response: Changed accordingly.

L34 - suggest adding another sentence on methods, to describe that all results and analysis are carried out through a normalization and comparison process to strain SR11 delta12.

Response: In the results section the normalization of all experiments analyzing STM adhesion to leafy green salad leaves to strain SR11 delta12 is well described. We do not see the need for highlighting this aspect in the abstract section.

L40 - this first sentence of the importance is not necessary, transmission of Salmonella through non animal sources is a well established fact.

Response: Sentence deleted.

L44 - what has to be part of further research does not describe what the importance of this work is, delete" and have to be part of further research"

Response: Changed accordingly.

L44-46 - this section should describe why it is important to identify the adhesive structures involved in binding to lettuce leaves. This sentence should be added to the abstract as it forms a more concise summary of the work than is described there.

Response: We revised Abstract and Importance sections accordingly

L54 - reference 2 appears to be an EFSA opinion paper, are there other references to describe the increasing number of Salmonella associated with Fresh produce?

Response: We added another reference.

L55 - i.e. examples of animal products and fresh produce are not necessary.

Response: Changed accordingly.

L57 - Is there a reference for seeds contamination leading to any outbreak from fresh produce other than sprouts? Pre-harvest factors for contamination expand well beyond just irrigation, seeds and fertilizers to all agricultural water use, intrusion by domestic and wild animals, employees,

equipment and tools used. Post harvest contamination includes factors beyond improper hygiene and washing/cutting. Suggest expanding and adding a reference.

Response: We added further references.

L60 - please cite directly an example where Salmonella was found to replicate on the surface of a fruit or vegetable under non-laboratory conditions. Is replication necessary for further spread?

Response: We cited a further study and review. We did not find studies on replication on fruits or vegetables under open field conditions but at least under laboratory conditions trying to mimic normal agricultural practices.

L62-69 - see above comment re. salad. I believe as written in this section that the term salad is specifically refereeing to a leafy green salad. I suggest being specific here that you are referencing leaf species, not all salad types, as salad's general meaning is a mixture of raw vegetables. This comment holds true throughout the paper.

Response: We changed the designation "salad" to "leafy green salad" throughout the paper.

L70 - please add the references to these studies investigating other adhesion mechanisms.

Response: We rewrote the sentence.

L73 - please specify that in this case internalization is through stomata, not through the general leaf surface as is implied.

Response: Changed accordingly.

L78 - 117 - this section describing adhesion mechanisms related to different structures should be significantly shortened to give just important highlights necessary to frame the work being done here; a further description of critical factors found to be important here can be included in the discussion.

Response: We have shortened the section.

L118- this section describing the Tet-On system is not necessary in the introduction. It could move to the methods if you wish to include.

Response: We shortened the paragraph.

L159 - unclear from my reading of the methods why more than 11 samples could not be performed at one time

Response: Preparation and handling of the lettuce leaf discs takes a certain amount of time. If more than 11 samples are handled at the same time, the lettuce leaf discs will be affected (e.g. degeneration will start since leaves are separated from the plant itself).

L166 - cognate lignands maybe absent on lettuce leaves - this was not mentioned/expanded upon in

the discussion. Are there references to further this finding?

Response: We added a section to the discussion.

L205-207 - this should move to the discussion

Response: We found this information better placed in the introduction.

L221-231 - this should move to methods.

Response: The information given here does not correspond to a methodological description, but explains the structure of the respective adhesin, which is important for the understanding of the following results concerning truncated forms of the adhesin.

L245-248 - this should move to discussion and methods.

Response: Without the explanations given here, we believe it is unlikely that the reader will understand the results that follow.

L287/fig 7 - it is unclear what the purpose of Figure 7 is. This is its only reference. Please expand on why it is here, or reference it more in the text.

Response: Figure 7 only serves as a visual summary of the results obtained in this work thus it is only mentioned one time. The summary serves to facilitate the comparison of the collected results regarding lettuce leaves in this study, but also the comparison to the previous study on the adherence factors of STM on lamb's lettuce leaves (Elpers et al 2020).

L301-304 - I can understand why you are hypothesing this direct motility, but I think it should be framed more as a hypothesis, as you did not do direct microscopy to determine if infact this is what was happening on your samples.

Response: Changed accordingly.

L309 - how long does it take to do this swimming and searching? How far is it possible for Salmonella to swim? How would the density of preferred sites on the leaves impact the attachment? Are there articles that reference the density of preferred sites on different types of leafy greens?

Response: Citations added for studies that support such role in interaction with mammalian cell.

L314-315 - suggest deleting this reference to virulence in mammalian cells. Salmonella is not virulent to plant cells, so this adds nothing to the findings.

Response: We have rewritten the section.

L321-322 - linking studies on Arabidopsis and Romaine attachment to its important on "all plants", which implies all plant tissues, is a bit of a stretch. Consider deleting or softening the language here.

Response: Changed accordingly.

L381 - is the *S. enterica* lifestyle being spoken of here in mammalian cells? Does *S. enterica* have a lifestyle on plants?

Response: We rewrote the sentence. Studies showed that BapA can be involved in the lifestyle during infection of mammalian cells, as well as outside the host, e.g. during biofilm formation on abiotic surfaces.

L 415-416 - delete this last sentence.

Response: Changed accordingly.

Discussion in general - I think what might be missing here is something about the bigger picture of why this work is important - you start your introduction speaking about increasing outbreaks from produce, and multiple ways plants can be contaminated. Could you wrap this up by speaking back to these points, and how better understanding the initial attachment could help in preventing contamination?

Response: We included a further section in the discussion about possible anti-adhesive structures during agricultural growth.

L429 - do we know what the original sources of the *Salmonella* Typhimurium strains are? While beyond the scope of this work, I'm curious if the same results would hold true for recent outbreak strains of *Salmonella*.

Response: The here used *Salmonella enterica* serovar Typhimurium strain NCTC 12023 (also called ATCC 14028) was originally isolated from pooled heart and liver samples from 4-week old chickens in 1960 (<https://doi.org/10.1128/JB.01233-09>). We are aware that STM may have evolved. However, yet we do not know any further adhesive structures encoded by STM which might impact adhesion to leafy green salad.

L427 - How large are the leaf disks?

Response: The leaf discs had a diameter of 8 mm. We ensured that the leaf discs were not floating and prevented contact of bacteria with the cut edges by using inlays of stainless steel with an inner diameter of 6 mm placed on the leaf discs. Leaf discs with the distinct inlays ensured a repeatable area of leaf being colonized by specific amounts of bacteria. We added a further reference that describes the applied methods step by step with detailed illustration.

L476 - for all replicates you had exactly the same inoculum level? There was no variation in here. if you used a lower inoculation level, does this impact results?

Response: There was not variation here regarding the inoculum. We did not check for the impact of lower inoculation levels of STM on the adhesion to lettuce leaves.

L476-477 - did 50 ul cover the entire surface of the leaf disk. How did you ensure in this experimental design the attachment was to the intact surface of the leaf and not the cut edges of the leaf. If you

could not, how might attachment to cut surfaces be different from attachment to intact surfaces on the leaf, and how may this impact the interpretation of all your results.

Response: The leaf discs had a diameter of 8 mm. We ensured that the leaf discs were not floating and prevented contact of bacteria with the cut edges by using inlays of stainless steel with an inner diameter of 6 mm placed on the leaf discs. Leaf discs with the distinct inlays ensured a repeatable area of leaf being colonized by a specific amount of bacteria. Using 50 μ l of the inoculum covered the area of a leaf disc with 6 mm diameter. We added a further reference describing the applied methods step by step with detailed illustration.

L477 - why one hour? What are RT conditions in your laboratory in terms of temperature and relative humidity? How would temperature impact or change the results?

Response: 1 hour incubation allows the determination of adherent bacteria without affecting the plant material severely. Room temperature means 21 °C. Since inoculation occurred by bacteria in buffer, the relative humidity appears of limited relevance of this assay. We tried once to affect STM adhesion to leaf green leaves by changing the temperature. This resulted in decreased adhesion at lower temperatures.

L478 - how were leaf disks washed? How much washing matrix was used? About how long was short mixing on the vortex?

Response: Leaf discs were washed once with 200 μ l PBS when inlays were still placed on the leaf discs. Then after collection of leaf discs in tubes, leaf discs were washed twice with 1 ml PBS. Short mixing on the vortex equals 2 seconds. We added a further reference describing the used method step by step with detailed illustration.

L484 - 488 - do you ever report the actual concentrations in your samples? I'm curious to what % actually attach, when not normalized. In general how representative is this inoculation procedure when compared to how Salmonella may be contaminating plants in the field? How could this inoculation style impact your results when compared to natural contamination?

Response: In the Materials and Method section we report using 50 μ l of an inoculum of 5.6×10^7 bacteria/ml. The supplementary figure 1A shows adhesions rates in % of the inoculum for STM to corn salad leaves and to lettuce leaves. Therefore, the actual adhered number of bacteria to the leaf disc can be calculated easily. Here we do not want to investigate the possible routes of contamination, but rather the influence of the various adhesive structures for binding STM to leaves of leafy green salads. It is obvious that the inoculation style has an influence on the further colonisation of STM. Natural contamination conditions include an enormous number of possibilities (irrigation, temperature, UV-light, microbiome, "media" in which the STM are present; ...) which could only be tested to a limited extent in this experimental set-up.

November 29, 2022

Prof. Michael Hensel
Universität Osnabrück
Abt. Mikrobiologie
Barbarastr. 11
Osnabrück 49076
Germany

Re: Spectrum03436-22R2 (Factors required for adhesion of *Salmonella enterica* serovar Typhimurium to *Lactuca sativa* (lettuce))

Dear Prof. Michael Hensel:

Your manuscript has been accepted, and I am forwarding it to the ASM Journals Department for publication. You will be notified when your proofs are ready to be viewed.

Sincerely,

Sandeep Tamber
Editor, Microbiology Spectrum
